# Sensory neurons regulate stimulus-dependent humoral immunity in mouse models of bacterial infection and asthma

Diane Aguilar[1,2,12], Fengli Zhu[1,2,12], Antoine Millet[1,2], Nicolas Millet [2,3], Patrizia Germano[4,5,6], Joseph Pisegna[5,7,8], Omid Akbari [9], Taylor A. Doherty[10], Marc Swidergall [2,3,11] & Nicholas Jendzjowsky [1,2,11] ✉

Sensory neurons sense pathogenic infiltration to drive innate immune responses, but their role in humoral immunity is unclear. Here, using mouse models of *Streptococcus pneumoniae* infection and *Alternaria alternata* asthma, we show that sensory neurons are required for B cell recruitment and antibody production. In response to *S. pneumoniae*, sensory neuron depletion increases bacterial burden and reduces B cell numbers, IgG release, and neutrophil stimulation. Meanwhile, during *A. alternata*-induced airway inflammation, sensory neuron depletion decreases B cell population sizes, IgE levels, and asthmatic characteristics. Mechanistically, during bacterial infection, sensory neurons preferentially release vasoactive intestinal polypeptide (VIP). In response to asthma, sensory neurons release substance P. Administration of VIP into sensory neuron-depleted mice suppresses bacterial burden, while VIPR1 deficiency increases infection. Similarly, exogenous substance P delivery aggravates asthma in sensory neuron-depleted mice, while substance P deficiency ameliorates asthma. Our data, thus demonstrate that sensory neurons release select neuropeptides which target B cells dependent on the immunogen.

The lung risks exposure to atmospheric infectious agents and allergens with each breath. Sensory neurons work alongside immune cells and influence host defense against infection and injury[1–7] but appear to exacerbate allergy[8–12]. Stimulation of sensory neurons by cytokines[4,5,8,12,13], proteases[11,14–17], PAMPs[1–3] and immunoglobulins[9,18–20] lead to neuronal depolarization. The release of neuropeptides by sensory neurons in response to infection or allergy[21–23], is a necessary cofactor for immune coordination[21,24]. The unique interplay between neurons and immune cells is pathology-dependent. A growing body of work demonstrates immune-stimulus-dependent neuropeptide release and, therefore, differential coordination of varying cell types[1–3,7,25–28]. Such differences speak to the dichotomous neuro-immune response, which can elicit a pro- or anti-inflammatory state[21,29].

[1]Division of Respiratory and Critical Care Medicine and Physiology, Harbor-UCLA Medical Center, Torrance, CA, USA. [2]The Lundquist Institute for Biomedical Innovation at Harbor-UCLA Medical Center, Torrance, CA, USA. [3]Division of Infectious Disease, Harbor-UCLA Medical Center, Torrance, CA, USA. [4]Research Service, Veterans Affairs Greater Los Angeles Healthcare System, Los Angeles, CA, USA. [5]CURE/Digestive Diseases Research Center, Department of Medicine, University of California, Los Angeles, CA, USA. [6]Division of Human Genetics, David Geffen School of Medicine at UCLA, Los Angeles, CA, USA. [7]Division of Gastroenterology, Hepatology and Parenteral Nutrition, VA Greater Los Angeles Healthcare System and Department of Medicine, Los Angeles, CA, USA. [8]Division of Pulmonary and Critical Care, Veterans Affairs Greater Los Angeles Healthcare System, Los Angeles, CA, USA. [9]Department of Molecular Microbiology and Immunology, Keck School of Medicine, University of Southern California, Los Angeles, CA, USA. [10]Division of Allergy and Immunology, Department of Medicine, University of California San Diego, Veterans Affairs San Diego Healthcare System, La Jolla, CA, USA. [11]David Geffen School of Medicine, Los Angeles, CA, USA. [12]These authors contributed equally: Diane Aguilar, Fengli Zhu. ✉e-mail: nicholas.jendzjowsky@lundquist.org

Most current evidence involving sensory neuron and immune cell crosstalk involves cellular-mediated immunity. For example, sensory neurons inhibit bacterial clearance and fungal infection by suppressing Th1 and Th17-mediated cytokine release and, therefore, suppress neutrophil-mediated clearance of infection in response to acute infection in naïve mice[2,7]. Given that many infectious diseases are recurring, prolonged, or stem from commensal microbes becoming invasive[30,31], further inquiry into the neural influence on humoral immunity in response to infection is warranted.

Conversely, during allergy and asthma, vasoactive intestinal polypeptide (VIP)[8] and substance P[11] release, activate ILC2s[8] and dendritic cells[11] to increase eosinophilic infiltration[8] and mast cell degranulation[11]. However, the catalytic event that drives allergic symptoms[32] is a dysregulated increase of allergen-specific IgE, which affects downstream targets. Given the very high prevalence of allergic disorders, there is a great need to further understand the role of sensory neurons in regulating humoral immunity.

Early investigations have shown that B cells respond to exogenous neuropeptides in vitro and can steer the predominant release of specific immunoglobulin[33–37]. For example, B cells express neurotransmitter receptors for, substance P[33], VIP[38,39], noradrenaline, and neuropeptide Y (NPY)[40,41]. In vitro, some of these neurotransmitters appear to increase immunoglobulin release[33–37]. Further, sensory neurons promote B cell class switching in the spleen[42] and lung[10]. Previously, it was shown that sensory neuron ablation reduced IgE release subsequent to a reduction of B220+ cells in the lung in response to house dust mite asthma and calcipotriol skin allergy[10]. Substance P was demonstrated to be responsible for IgE class switch recombination in vitro, and recent in vivo results support substance P-mediated class switch[10].

Here, we show that the release of neuropeptides by sensory neurons differs between our models of *S. pneumoniae* and *A. alternata*-induced asthma. Sensory neuron depletion suppresses sensory neuropeptides in the lungs of each model and, therefore, reduces pulmonary B cell and plasma cell immunoglobulin release; experiments using knockout mice for respective neuropeptides or receptors support neuron depletion data. Supplementation of neuropeptides back into sensory neuron-depleted mice in each model either improves bacterial clearance or worsens asthma features due to a restoration of immunoglobulin production and release- thus supporting the sufficiency of sensory neurons to regulate humoral immunity. In summary, the release of sensory neuropeptides is altered by the immune stimulus, which appears to provide a way to steer the humoral response.

## Results

### TRPV1+ neurons mediate survival and bacterial clearance in pneumonia

Transient receptor potential vanilloid 1+ (TRPV1) neurons have been demonstrated to influence T cell-mediated coordination of immunity by the release of specific neuropeptides[2,3,8]. The TRPV1 ion channel responds to capsaicin, protons, prostaglandins, lipids, and heat stimuli[43]. TRPV1+ neurons have been found to coordinate cellular immunity and influence T cell stimulation of neutrophils in acute *Staphylococcus aureus* infection[2] and eosinophils and mast cells in allergic asthma[8]. Thus, we tested how TRPV1+ sensory neurons play a role in humoral immunity.

We used an established model of sensory neuron chemical ablation with resiniferatoxin (RTX)[2,44,45], we suppressed TRPV1-containing neurons similarly in the vagal (nodose/jugular) ganglion by >85% compared to *TRPV1-DTR* mice injected with the diphtheria toxin directly into the vagal ganglia as assessed by qPCR and TRPV1 immunohistochemistry (Supplementary Fig. 1). RTX, administered by subcutaneous injection, induces global ablation of TRPV1+ neurons, thus blocking sensory neuron vesicle emission of neuropeptides and rendering them ineffective/atrophied[2,44,45]; we confirmed the suppression

of neuropeptide transcripts in the vagal ganglia with RTX by qPCR (Supplementary Fig. 1). Although there are indications that dorsal root ganglia (distal site, primarily innervate skin and visceral organs[46]) provide some innervation of the lungs[47], recent tracer studies have demonstrated sparse innervation with an overwhelming majority of lung sensory neuron innervation provided by the vagal ganglia[48].

To investigate humoral immunity in response to *S. pneumoniae* lung infection, we aimed to replicate the colonization and infection patterns observed in humans, considering that mice are not naturally colonized with *S. pneumoniae*[49,50]. We established a model of *S. pneumoniae* infection, which involved exposure of mice to a low inoculum on Day 0 and subsequent exposure to a high dose of bacteria on Day 9 (Fig. 1a). Following the low inoculating dose, both sensory neuron intact, and RTX mice showed a similar amount of bacteria remaining after the low inoculum (Fig. 1b). Both groups were able to clear the bacteria prior to the infectious dose (Fig. 1b). Compared to naïve mice, our *S. pneumoniae* model showed an initially increased bacterial burden, which was not different from RTX mice, 16 h after infection (Fig. 1b). Forty-eight hours after the last infectious dose of *S. pneumoniae*, sensory neuron intact mice almost completely cleared bacteria. In contrast, RTX-treated mice exhibited an elevated bacterial burden compared to sensory intact mice and naïve mice, which persisted six days after the final infection (Fig. 1b). We further assessed the role of sensory neurons in cross-protection against a lethal pneumococcal strain. Sensory neuron ablation resulted in a higher bacterial burden and reduced survival when pre-exposed to serotype 19 F and infected with serotype 3 (Fig. 1c, d). In addition, in response to a single high dose of *S. pneumoniae*, in both sensory neuron intact and RTX ablated mice, bacterial burden and IgG remained unchanged, and when exposed to a single infectious dose of serotype 3, survival was unchanged (Supplementary Fig. 2); similar to previous investigations[2]. Bacterial clearance between RTX-treated and targeted vagal TRPV1 ablation with direct diphtheria toxin (DTX) injection into the vagal ganglia of *TRPV1-DTR* mice[51] was similar (Supplementary Fig. 3). Consequently, our findings suggest that vagal-specific sensory neurons are required to control *S. pneumoniae* during lung infection in our recall model.

Next, we determined whether nociceptors regulated pro-inflammatory-cytokine production before lung infection clearance. Post-infection, RTX-treated mice showed reduced inflammatory cytokine levels, including IL-1β, IL-6, and TNF (Fig. 1e and Supplementary Fig. 4). Furthermore, cytokines involved in B cell stimulation (BAFF, TNFSF13, IL-2, and IL-5) and chemokines involved in B cell attraction (CXCL13, CCL5, and CCL7) were reduced in RTX mice compared to control mice (Fig. 1e and Supplementary Fig. 4).

### TRPV1+ neurons regulate B cells

Given that cytokines and chemokines for B cell recruitment (Fig. 1e) were reduced, we assessed immunoglobulin levels in lung homogenates. Immunoglobulins were reduced in RTX mice compared to sensory neuron intact mice after 16 h of infection (Fig. 2b). *S. pneumoniae* serotype 19 F specific IgG was also reduced in RTX as the IC50 calculated from lung homogenate dilutions was higher in RTX compared to Veh (Fig. 2b). The reduction in immunoglobulins was directly due to a reduction in isotype-switched memory, resident memory B cells, plasma cells, and plasmablasts (Fig. 2c–g and Supplementary Figs. 5–7). It is important to note that RTX did not affect B cell populations immediately before pre-exposure or after first exposure (Fig. 2c–g). The reduction in B cell populations remained significant at 48 h (Fig. 2c–g and Supplementary Fig. 6), in direct accord with an augmented bacterial burden in RTX mice (Fig. 1b, c). As B cells were diminished in RTX following infection, in agreement with immunoglobulins, our data were consistent with a role for sensory neurons with adaptive immunity. In contrast, T cells were unaffected by sensory neuron ablation (Supplementary Fig. 8).

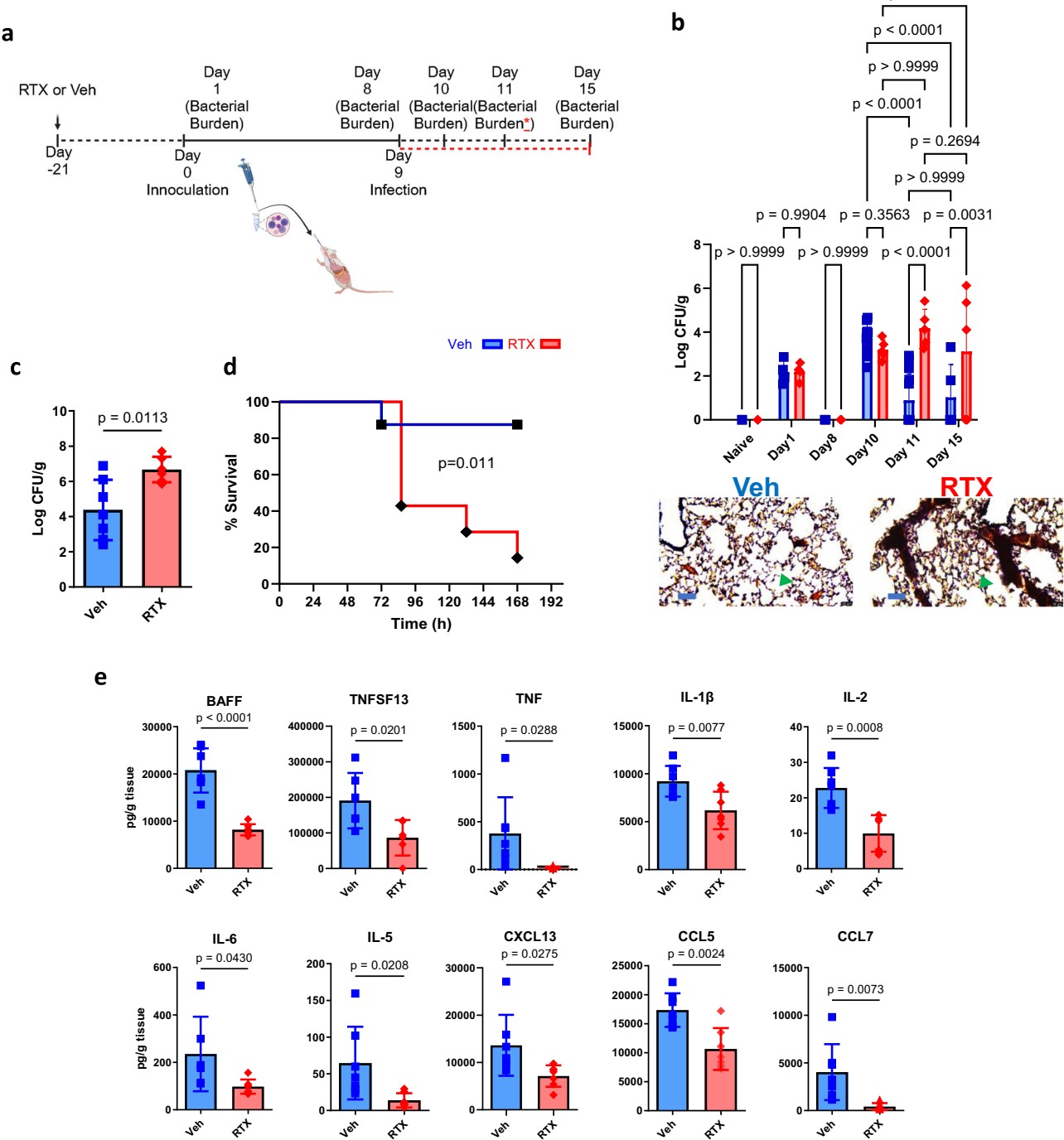

**Fig. 1 | Sensory neurons are required for successful clearance of *S. pneumoniae* following pre-exposure and infection. a** Model of *S. pneumoniae* pre-exposure and infection, created in BioRender. Aguilar, D. (2022) BioRender.com/h64o602. Mice received escalating doses of resiniferatoxin (RTX) or Vehicle (Veh) starting 21 days prior to inoculation. $10^4$ CFU of *S. pneumoniae (Serotype 19 F ATCC49619)* in 50 μl PBS was delivered on day 0 to expose mice, then $10^8$ CFU in 50 μl PBS was delivered on Day 9. Bacterial burden was enumerated on Days 10, 11 and 15. The red asterisk and dotted line denote the separate set of experiments assessing cross-protection between serotypes: $10^4$ CFU of *S. pneumoniae (Serotype 19 F ATCC49619)* was delivered on day 0 to expose mice, then $10^6$ CFU *(Serotype 3, ATCC6303)* in 50 μl PBS was delivered on Day 9, survival was assessed as was bacterial burden on Day 11. **b** Bacterial recovery from lungs in untreated sensory neuron intact mice (Naïve), untreated sensory neuron ablated mice (Naive RTX), pre-exposed and infected sensory neuron intact (Veh), or pre-exposed and infected sensory neuron ablated mice (RTX). Naïve $n = 6$, Naïve RTX: $n = 5$; Day 1: Veh: $n = 4$, RTX: $n = 4$; Day 8: Veh:

$n = 5$, RTX: $n = 5$; Day 10: Veh: $n = 13$, RTX: $n = 5$; Day 11: Veh: $n = 13$, RTX: $n = 5$; Day 15: Veh: $n = 5$, RTX: $n = 5$. Two-way ANOVA with Newman-Keuls post hoc test, mean ± sd. Data were pooled from three independent experiments. Gram stain shows *S. pneumoniae* infection (purple and green arrows point to example of gram-positive space) was greater in RTX compared to Veh. Scale bar = 20 μm. **c** Veh mice had reduced bacterial burden in response to primary inoculation with serotype 19 F and infection with serotype 3 (Veh $n = 7$, RTX $n = 6$). Data were compared with a two-sided *t* test, mean ± sd. **d** 90% of Veh mice survived infection with serotype 3 compared to 20% of RTX mice after inoculation with 19 F (Veh $n = 8$, RTX $n = 7$). Data were compared with the Log-rank (Mantel-Cox) test, mean ± sd.
**e** Quantification of B cell activating factor (BAFF), APRIL13 (TNFSF13), TNF, IL-1β, IL-2, IL-6, IL-5, CXCL13, CCL5, CCL7 in sensory neuron intact (Veh $n = 7$) and sensory neuron ablated (RTX, $n = 6$) mice 16 h following the final $10^8$ CFU dose of *19 F*. Samples were analyzed with Luminex bead assay. IL-6 was re-run with ELISA. Two-sided *t* test, mean ± sd. Data were pooled from 2 independent experiments.

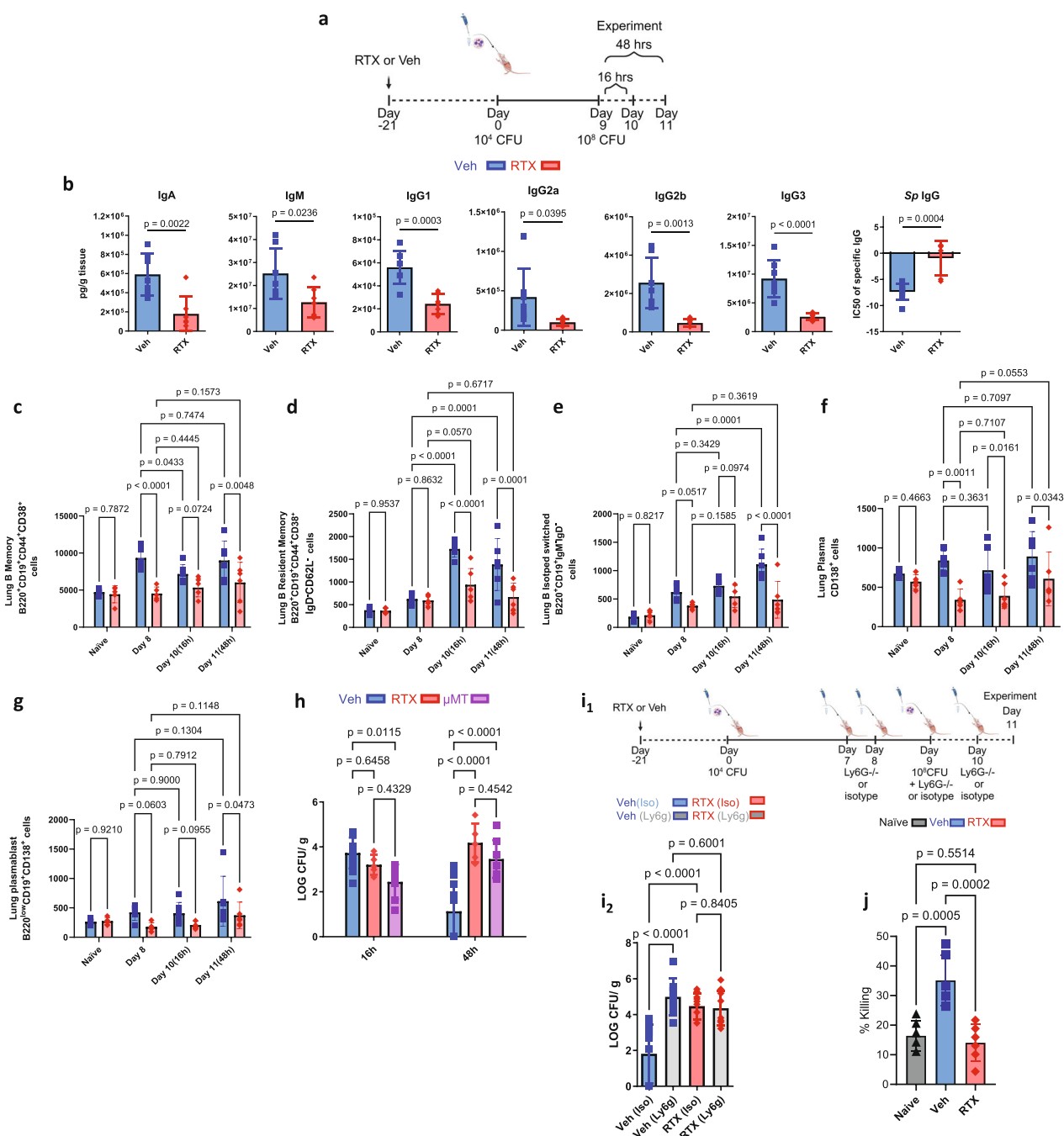

To confirm that bacterial clearance in response to infection after pre-exposure relied on intact B cells, we compared bacterial burden from our sensory neuron intact and depleted mice to mice lacking mature B cells (µMT, *Ighm*⁻/⁻ mice). Like RTX mice, µMT mice had a significantly increased bacterial burden at 48 h after our recall model of *S. pneumoniae* (Fig. 2h).

Neutrophil depletion in sensory neuron intact mice significantly increased the bacterial burden to a level similar to RTX mice (Fig. 2i and Supplementary Fig. 9). Neutrophil depletion did not further augment bacterial burden in RTX mice, showing that the interaction between B cells and neutrophils may be an important mechanism of bacterial clearance in our recall model (Fig. 2i and Supplementary Fig. 9). To confirm the B cell-neutrophil axis, naïve neutrophils were incubated with decomplemented serum harvested from sensory neuron intact and RTX ablated mice, which underwent our pre-exposure and infection model, as well as naïve mice. Neutrophils that were

incubated with serum from sensory neuron intact mice had the highest level of bacterial killing (Fig. 2j). Therefore, the main effect of sensory neurons in response to infection was to coordinate B cell homing and immunoglobulin production to enhance neutrophil-mediated *S. pneumoniae* clearance.

## TRPV1+ neuron neuropeptide release of VIP stimulates B cell immunoglobulin release

B cells express select neuropeptide receptors[52]. Therefore, we reasoned that neurons would stimulate B cells within their proximity. Using a sensory neuron reporter mouse (*Vglutcre-tdTomato*[48]), we showed that B220⁺ cells were localized within <20 µm of sensory neurons in the lungs 48 h after the final infectious dose of *S. pneumoniae* (Fig. 3a). In comparison, infected mice treated with RTX, showed a more dispersed distribution of B220⁺ cells, extending further from sensory neurons. Of note, whereas RTX depletes TRPV1+

**Fig. 2 | Sensory neuron ablation reduces immunoglobulins, B cells, and disrupts neutrophil effector function after *S. pneumoniae*. a** In response to our pre-exposure and infection model (as in Fig. 1a), the image created in BioRender. Aguilar, D. (2022) BioRender.com/h64o602. **b** IgA, IgM, IgG1, IgG2a, IgG2b, and IgG3 were reduced with sensory neuron ablation 16 h (Day 10) after the final infection. Veh: $n = 7$, RTX: $n = 7$. Isotyping Luminex was used to analyze lung homogenates. Two-sided $T$ test, mean ± sd. IgG specific to *S. pneumoniae* was tested with crude *Sp* extract. Serial dilutions were used to model the IC50 with a 4-point log model to determine differences between Veh ($n = 8$) and RTX ($n = 6$). Two-sided $T$ test, mean ± sd. Data were collected from two independent experiments. **c** B memory B220$^+$CD19$^+$CD44$^+$CD38$^+$ cells, **d** B resident memory B220$^+$CD19$^+$CD44$^+$CD38$^+$IgD$^-$CD62l$^-$ cells, (**e**) Isotype switched B220$^+$CD19$^+$IgM$^-$IgD$^-$ cells, (**f**) Plasma CD138$^+$ cells and, (**g**) Plasmablast CD138$^+$B220$^{low}$CD19$^-$ cells were downregulated by sensory neuron ablation at 16 and 48 h after infection. Only B memory and plasma cells were different between Veh and RTX on Day 8. Two-way ANOVA with Holm-Sidak test, mean ± sd. Day 0: Veh: $n = 5$, RTX: $n = 5$; Day 8: Veh: $n = 5$, RTX: $n = 5$; Day 10: Veh: $n = 6$, RTX: $n = 6$; Day 11: Veh: $n = 6$, RTX: $n = 6$. **h** µMT mice had reduced burden at 16 h in comparison to sensory neuron intact mice (Veh). Sensory neuron ablated (RTX) mice demonstrated a bacterial burden similar

to mice with immature B cells (µMT) 48 h after infection. Sensory neuron intact mice (Veh) reduced their bacterial burden at 48 h. Veh $n = 13$, RTX $n = 5$, µMT infected $n = 7$. Data from four independent experiments. Two-way ANOVA with Holm-Sidak test, mean ± sd. **i** Ly6G A18 neutrophil depletion antibody or isotype control was delivered before, during, and after the final infection dose of *S. pneumoniae;* the image was created in BioRender. Aguilar, D. (2022) BioRender.com/h64o602. Bacterial burden (Log CFU/g lung mass) from lung homogenates was greater with neutrophil depletion in sensory neuron intact infected mice compared to isotype control 48 h (Day 11) after infection. Sensory neuron-depleted mice (RTX) did not further increase bacterial burden with neutrophil depletion, demonstrating that B-cell neutrophil interaction was disrupted. One-way ANOVA with Holm-Sidak post-hoc test, mean ± sd. Veh (Iso): $n = 12$, Veh (Ly6g): $n = 8$, RTX (iso): $n = 10$, RTX(Ly6g): $n = 9$. Data were pooled from two independent experiments. **j** Naïve neutrophils were incubated with decomplemented serum harvested from naïve, Veh, or RTX mice (pre-exposed and infected) and plated in *S. pneumoniae* 19 F coated wells (1000 CFU). Neutrophils were lysed 30 min later and plated on blood agar plates, and CFU was enumerated 24 h later. One-way ANOVA with Holm-Sidak post-hoc test, mean ± sd. $n = 6$ per group.

---

neurons, sensory neurons have a heterogeneous genetic composition[48,53], and therefore, some sensory nerves remain following *TRPV1* depletion (Fig. 3a). Thus, it was likely that RTX suppressed sensory neuropeptides released by TRPV1+ neurons. Therefore, we measured known sensory neurotransmitters with noted immunological effects (calcitonin gene-related peptide (CGRP)[2], Substance P[10,11,33,54,55], VIP[8,36,39,56], NPY[34,41,57]) from lung homogenates. We showed pre-exposure and infection increased NPY and VIP (Fig. 3b, c). Compared to sensory neuron intact mice, we demonstrated that RTX suppressed NPY, VIP, and CGRP (Fig. 3d, e). The suppressed neuropeptide concentrations in the lungs were associated with reduced cDNA expression in the vagal ganglia (Supplementary Fig. 1). The neuropeptide concentrations measured from the lungs were inversely correlated with bacterial burden and positively correlated with immunoglobulins (Supplementary Fig. 10). Therefore, we hypothesized that NPY or VIP released from sensory neurons were able to increase B cell immunoglobulin release, to stimulate neutrophil-mediated bacterial clearance. Neuropeptides bind to specific receptors. We show that VIPR1 and NPY1R were both present in B cells and plasma cells, which demonstrated the likelihood of stimulation by sensory neuropeptides (Supplementary Fig. 11).

We isolated splenic B cells and cultured them with NPY or VIP. VIP stimulation increased IgG secretion (Fig. 3i) and surface-bound levels in CD138$^+$ plasma cells (Fig. 3f, g and Supplementary Fig. 12). However, neurotransmitter stimulation did not affect the proliferation of B cells and plasma cells, as assessed by Ki67$^+$ intracellular staining (Fig. 3h). Given the high expression of VIPR1 on B cells, we tested the effects of VIP stimulation in WT and *VIPR1$^{-/-}$* isolated B cells. In *VIPR1$^{-/-}$* cultured B cells, VIP and other neuropeptides did not increase bound and released IgG. This demonstrated that VIPR1 was the main receptor responding to VIP stimulation (Fig. 3j–l). We then isolated B cells from the lungs of naïve, and pre-exposed and infected mice. VIP again increased bound and released IgG (Fig. 3m–o). However, we noted that pre-exposed and infected mice had a suppressed response to all conditions. Therefore, we tested whether this suppression was due to B cell exhaustion. B cells harvested from pre-exposed and infected mice, which were then re-stimulated in culture for 96 h, showed an augmentation of CD11c and reduction of CD23, to a greater extent than lung B cells harvested and cultured from naïve mice (Fig. 3p, q), indicative of B cell exhaustion[58–60].

We supplemented sensory neuron intact and RTX sensory-depleted mice with VIP to test the functional outcome of VIP release in response to *S. pneumoniae* (Fig. 4a and Supplementary Fig. 13). VIP supplementation rescued RTX mice by suppressing their bacterial burden at 48 h post-infection (Fig. 4n). Furthermore, B memory,

plasma cells, plasmablasts, and lung IgG were increased with VIP supplementation in RTX mice compared to RTX mice supplemented with PBS (Fig. 4b–g). Furthermore, *VIPR1$^{-/-}$* had an altered accumulation of select B cell populations (Fig. 4h–l), reduced immunoglobulins and cytokines (Fig. 4m and Supplementary Fig. 14), and increased bacterial burden in comparison to WT littermates in response to pre-exposure and infection (Fig. 4o). In addition, when neutrophils were ablated with a neutralizing antibody, *VIPR1$^{-/-}$* mice did not further increase the bacterial burden in their lungs following pre-exposure and infection (Fig. 4p). These data showed that VIP stimulation of B cells through VIPR1 increased B cell-neutrophil interaction through an increased release of immunoglobulins.

In order to rule out non-neuronal sources of VIP, we stained for VIP in T cells, macrophages, and neutrophils, which others have identified as potential non-neuronal sources of VIP[61–63]. RTX did not suppress VIP expression in leukocytes (Supplementary Fig. 15). Neuroendocrine cells within the lungs also have the potential to release neuropeptides[64]. However, VIP fluorescence was not diminished with RTX in the epithelium where pulmonary neuroendocrine cells reside (Supplementary Fig. 15). We note that in conjunction with the reduction of VIP in RTX lung homogenates (Fig. 3b), the suppressed expression of VIP transcripts in vagal ganglia in RTX compared to vehicle mice (Supplementary Fig. 1) and a maintained amount of VIP-positive lymphocytes and neuroendocrine cells, the predominant source of neuropeptides in response to *S. pneumoniae* was likely lung TRPV1$^+$ sensory neurons.

## Neural stimulation of immunoglobulins exacerbates asthma features

Given that immunoglobulins and humoral immunity are important mechanisms for bacterial clearance, we then asked whether an over-production of immunoglobulins as a result of neural stimulation could have resulted in immune dysfunction. We hypothesized that this would be neuropeptide-driven but, likely mediated by an alternate peptide, as previous data supports[10,55]. *A. alternata* caused a swift and distinct rise of IgE in response to four inoculations (Fig. 5a, b) as demonstrated previously[65]. *A. alternata*-induced asthma increased airway hyper-responsiveness and goblet cell metaplasia, which was reduced in sensory neuron-depleted mice (Fig. 6b–e), where airway hyper-responsiveness was correlated with IgE (Fig. 6c). Specific B cell recruitment of cytokines was suppressed with RTX sensory neuron ablation (Supplementary Fig. 16). RTX also suppressed B cells, immunoglobulins, and mast cells in asthmatic mice (Fig. 5c–h and Supplementary Figs. 5, 17). *A. alternata* specific IgE was also reduced in RTX mice as the IC50 calculated from lung homogenate dilutions was

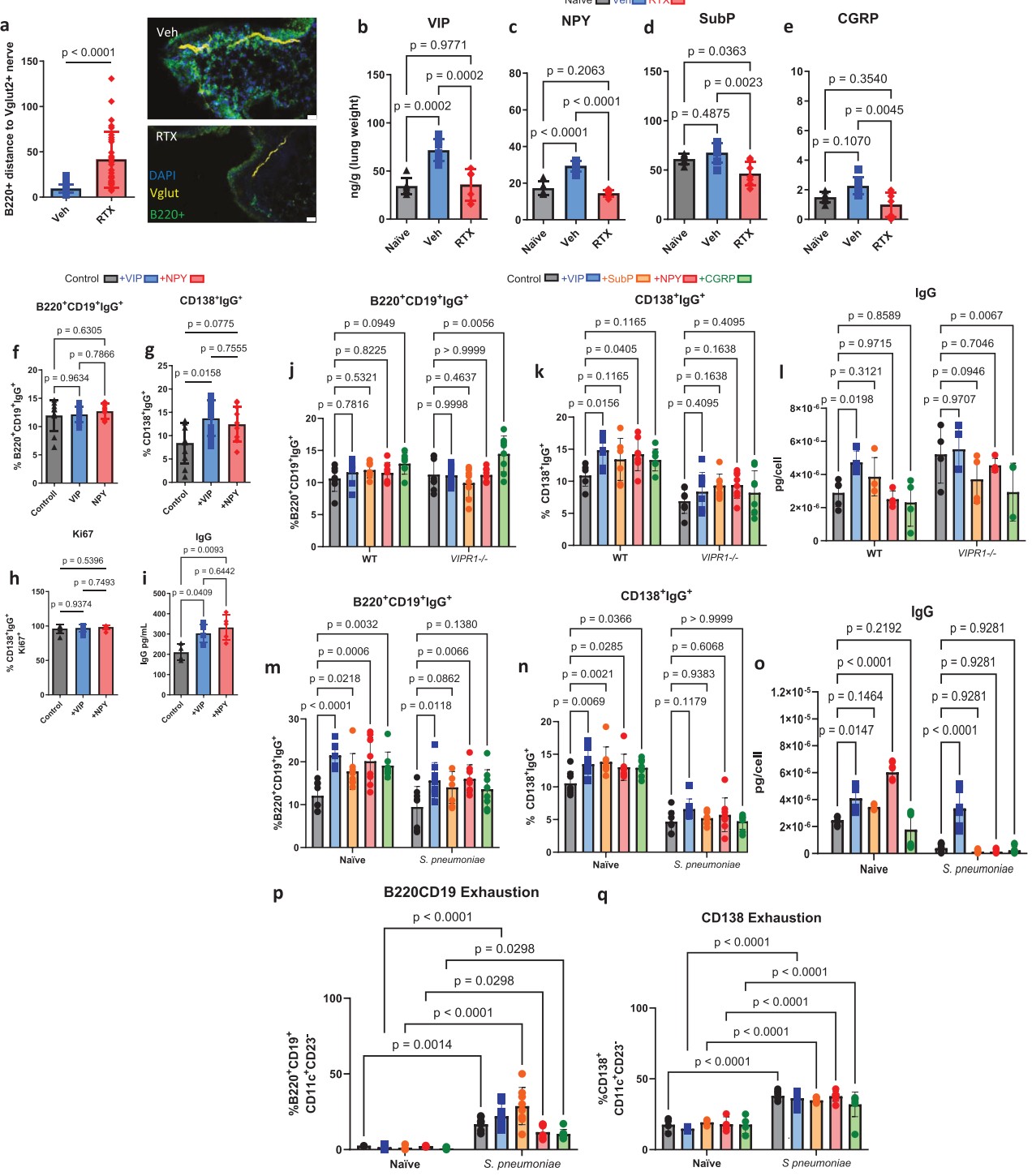

higher in RTX compared to Veh mice (Fig. 5b). NPY and Substance P were increased in the lungs from *A. alternata* mice, in comparison to naïve mice, while decreased with RTX (Fig. 7a–d). VIP was not increased in response to *A. alternata*-induced lung inflammation (Fig. 7d). Substance P receptors (Neurokinin 1 and 2) were not present in isolated naïve B cells (Supplementary Fig. 11), but cDNA was detected in lung B cells. Substance P also binds to mas-related g protein-coupled receptors[66], where cDNA and protein were present on all B cells (Supplementary Fig. 11). NPY and Substance P increased bound and released IgE in naïve splenic B cells (Fig. 7e–h and Supplementary Fig. 12). To test whether substance P could stimulate B cells, we supplemented substance P into *A. alternata* mice (Fig. 8a). We showed that

substance P supplementation increased B cell resident populations in the lungs of RTX sensory neuron-depleted mice treated with *A. alternata* (Fig. 8b–f). Further, IgE was increased when *A. alternata* RTX mice were supplemented with substance P (Fig. 8l). This was in line with the removal of substance P, as substance P knockout mice (*Tac1*[−/−], unable to produce substance P/ have receptors) showed reduced levels of IgE (Fig. 8m), lung-resident B cell populations (Fig. 8g–k), as well as cytokines and other immunoglobulins (Supplementary Fig. 18) in response to *A. alternata*-induced lung inflammation compared to WT mice with *A. alternata-induced* lung inflammation. To account for alternate sources of substance P, we stained for substance P in T cells, eosinophils, and mast cells, which could be sources of substance P[61–63].

**Fig. 3 | Vasoactive intestinal peptide attracts B cells and regulates immunoglobulin production, and release. a** B220$^+$ cells clustered around sensory nerves from the lungs of sensory neuron intact mice (Veh). In RTX sensory neuron-depleted mice, B220$^+$ cells are further dispersed. 30 cells (10 cells per sample) from $n = 3$ mice for Veh & RTX. Two-sided t-test, mean ± sd. **b–e** Prominent sensory neuron neuropeptides were increased in lung homogenates 16 h after pre-exposure and infection with *S. pneumoniae* compared to naïve mice. **b** VIP was increased with infection and suppressed with sensory neuron depletion. **c** NPY was increased with infection and suppressed with sensory neuron depletion. **d** Substance P did not significantly increase following infection with *S. pneumoniae*. **e** CGRP did not significantly increase following infection with *S. pneumoniae*. Lung homogenates were analyzed with ELISA. **b–e** PBS: $n = 6$, Veh: $n = 7$, RTX: $n = 7$. One-way ANOVA with Tukey's post hoc test, mean ± sd. Data from 2 independent experiments. **f–i** B cells were isolated from spleens of naïve mice and cultured in media with IL4 + LPS and supplemented with VIP or NPY. Cells and media were sampled after 96 h of incubation. **f** VIP did not increase IgG bound to B220$^+$CD19$^+$ cells. **g** VIP significantly increased bound IgG in CD138$^+$ plasma cells. **h** Proliferation, as assessed by intracellular Ki67 staining, was not affected by neuropeptides (from CD138$^+$ gate). $n = 10$

all conditions. **i** Secreted IgG was increased by VIP and NPY. Control: $n = 4$; VIP, NPY: $n = 5$. One-way ANOVA and Tukey's post-hoc test, mean ± sd. Data from 2 independent experiments. **j–l** B cells were isolated from the spleens of VIP1 receptor knockout (*VIPR1$^{-/-}$*) and WT littermates and cultured for 96 h. **j** B220$^+$CD19$^+$ cells were not significantly affected by VIPR1 deletion. **k** *VIPR1$^{-/-}$*CD138$^+$ cells did not significantly upregulate IgG in response to VIP and other neuropeptides. WT: $n = 7$; *VIPR1$^{-/-}$*: $n = 8$. **l** IgG released into the media was not increased with VIP or other peptides with VIPR1 genetic deletion, VIP increased WT B cell IgG release. WT: $n = 4$; *VIPR1$^{-/-}$*: $n = 4$. Two-way ANOVA with Holm-Sidak post-hoc test, mean ± sd. **m–p** B cells were isolated from the lungs of naïve or *S. pneumoniae* pre-exposed and infected mice. **m**, **n** VIP increased bound IgG in B220$^+$CD19$^+$and CD138$^+$ cells. This effect was greater in naïve cells. Naïve: $n = 8$, *S. pneumoniae*: $n = 8$. **o** released IgG was increased with VIP in both naïve and pre-exposed and infected cells. Naïve: $n = 4$, *S. pneumoniae*: $n = 5$. **p**, **q** pre-exposed and infected cells show higher markers of B cell exhaustion after additional stimulation in culture. Two-way ANOVA with Holm-Sidak post-hoc test, mean ± sd. Naïve: $n = 4$, *S. pneumoniae*: $n = 8$.

RTX did not suppress substance P expression in leukocytes (Supplementary Fig. 19). Substance P fluorescence was not diminished with RTX in the epithelium where neuroendocrine cells are located (Supplementary Fig. 19). In conjunction with the reduction of substance P in RTX lung homogenates (Supplementary Fig. 20), the suppressed expression of substance P transcripts in vagal ganglia in RTX compared to vehicle mice (Supplementary Fig. 1) and a maintained amount of substance P positive leukocytes and epithelium, we believe the predominant source of substance P in response to *A. alternata* were lung sensory nerves. In summary, our data demonstrate that substance P exacerbated asthmatic features in mice by increasing B cell production of IgE.

## Discussion

Models of repeated immune stimulation have shown that sensory neurons stimulate humoral immunity[10]. In contrast, sensory neurons appeared to suppress innate responses to select single-dose models of bacterial and fungal infection[1–7]. Recent data have shown that sensory neurons modulate B cell populations in the lungs[10]. Select neuropeptides, released by sensory neurons, induced IgE, IgG, and IgA release in vitro[33–37]. Therefore, we reasoned that sensory neurons could play critical roles in humoral immunity[10,42,67,68]. The differential roles of neurons with innate and adaptive immunity may explain why sensory neurons can produce divergent results depending on the underlying immune stimulus, which is likely dependent on the timing and type of neuropeptide released.

Sensory neurons sense pathogenic infiltration directly through PAMPs or indirectly by binding cytokines and proteases[21–23]. Upon stimulation, sensory neurons release neuropeptides directly into the immediate vicinity[21–23]. The type and amount of neuropeptide released is likely dependent on the neuron type stimulated, which would dictate how it senses the environment and which neuropeptides would be released[53,69]. Depletion of sensory neurons reduced B cell populations, IgG, and neutrophil activity, which subsequently augmented bacterial burden in our recall model. This humoral regulation was due to the release of VIP, likely from sensory neurons, and the resulting stimulation of B cells and plasma cells via the VIPR1 (Fig. 9). We demonstrated that B cells and plasma cells have VIPR1, which stimulate PKCs that are critical to the development and release of immunoglobulins[70,71]. These data were confirmed with *VIPR1$^{-/-}$* mice, highlighting the essential role of VIPR1 signaling in the proper functioning and immunoglobulin-secreting capability of B cells and plasma cells (Fig. 9). While RTX treatment suppressed neuropeptide gene transcripts in sensory neurons and reduced neuropeptide levels in lung homogenates, sensory neuropeptides may still have activated the release of neuropeptides from other sources, such as

lymphocytes and neuroendocrine cells. This potential alternative source of neuropeptide release means that our current analysis would have underestimated the overall stimulation of B cells by neuropeptides, underscoring the importance of neuropeptide stimulation of B cells.

Our results were consistent with early in vitro data which showed that VIP can stimulate B cell immunoglobulin release[33–37]. In the context of bacterial lung infection, our results contrast with the finding that sensory neurons suppressed immunity in response to acute infection[2]. Specifically, sensory neuron release of CGRP suppressed bacterial clearance of acute pulmonary *Staphylococcus aureus infection* by dampening γδT cell-neutrophil stimulation[2]. When we used a single dose of *S. pneumoniae*, we saw that sensory neuron depletion had a similar bacterial burden and IgG as sensory neuron intact mice 48 h following a single challenge as well as survival in response to a lethal strain. In response to pre-exposure and infection, CGRP did not significantly increase. We suggest that CGRP may be necessary for early innate neuroimmune interaction, which VIP then supersedes upon re-infection, or that CGRP may have an alternate role for B cell maturation in germinal centers[72]. While humans are colonized with pneumococcal strains at birth, mouse models appear to require pre-exposure prior to infection in order to engage humoral memory and mimic the human condition[30,31,49,50]. Therefore, our model of pre-exposure and infection may provide data that is more applicable to the translation to clinical practice.

B cells can become dysregulated and exacerbate diseases. In the context of allergen-induced lung inflammation, we showed that sensory neurons became hypersensitized and stimulated B cells. This was in direct accord with previous investigations demonstrating that hypersensitization of sensory neurons led to augmented cytokine release and subsequent eosinophil trafficking by T cells or innate resident lymphocytes[8,9,28]. Our study extended these findings by showing that sensory neuron depletion reduced B cell-attracting cytokines and, as a result, reduced B cell recruitment. Mast cells were also reduced in sensory neuron-depleted mice. It is likely that the combined reduction in B cells and IgE release also reduced mast cell stimulation. Conversely, mast cells also harbor neuropeptide receptors[73], and sensory neuron ablation could directly reduce mast cell stimulation.

Substance P released by sensory neurons regulates B cell stimulation in our asthma model. We and others show the ability of substance P to induce class switch recombination[10]. We demonstrate that mas-related gene protein-coupled receptors that are stimulated by substance P[11] are ubiquitously expressed by B cells (at the gene transcript and protein level), with tachykinin receptor transcripts being detectable in pulmonary B cells. Our data significantly extended

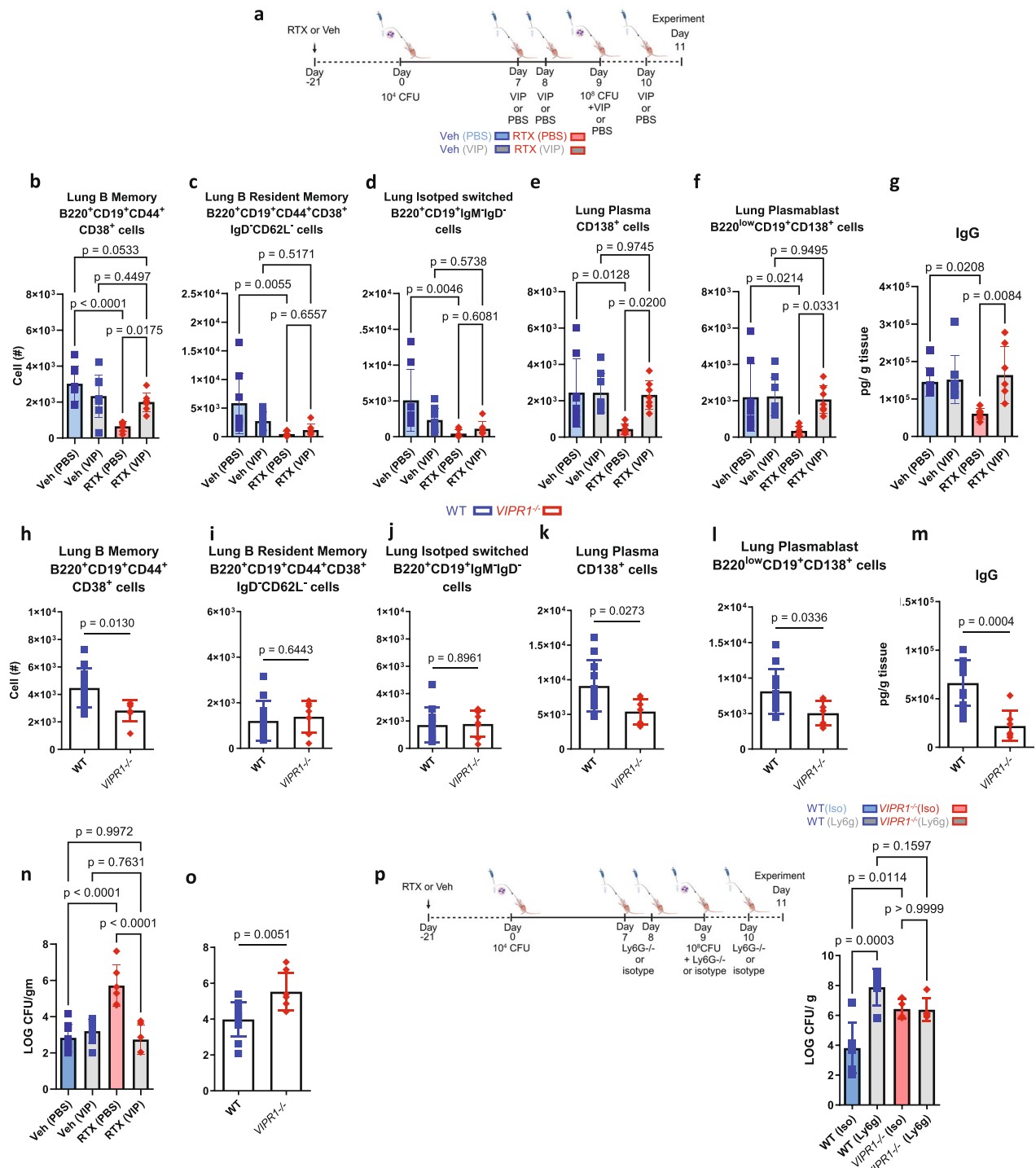

previous data by showing that the chemical ablation of sensory neurons reduced B cell populations and IgE levels. Further, when substance P was supplemented in asthmatic sensory neuron-depleted mice, IgE was once again restored. Finally, genetic ablation of the ability to produce substance P completely prevented IgE production in response to *A. alternata*. Increased substance P production has been associated with an asthmatic phenotype. Specifically, increased substance P-expressing neurons in asthma[74] augmented mucus secretion and goblet cell hyperplasia[75–77], which contributed to overall airway hyperresponsiveness[8], as we showed. Tachykinins such as substance P, therefore, contribute to asthma induction via direct airway stimulation and through IgE release (Fig. 9).

While the direct effects of neuropeptides on B cells have been demonstrated, recent findings show that the lymph node innervation by sensory neurons is restricted to the macrophage-rich medulla, not the lymphocyte-rich cortex[67,72,78]. Therefore, sensory neurons or neural circuits are unlikely to recruit B cells from lymph nodes or secondary lymphoid tissues. In line with this reasoning, we showed no changes in B cells in the spleens or bone marrow in response to sensory neuron depletion. Our data demonstrated that the effects of sensory neuropeptides are confined to the lungs and produce these effects locally. Lung tissue-resident B cells establish themselves after primary infection or colonization[79–82] and are poised to release antibodies[81,82] upon antigen encounter with a challenge infection. Our data demonstrated

**Fig. 4 | Vasoactive intestinal peptide supplementation improves bacterial clearance, memory B cells, plasma cells, and IgG in response to pre-exposure and infection with *S. pneumoniae*. a** Vasoactive intestinal peptide or vehicle (PBS) was delivered as per the schematic in addition to the model of *S. pneumoniae* pre-exposure and infection with serotype 19 F, image created in BioRender. Aguilar, D. (2022) BioRender.com/h64o602. Cells were gated on Live CD45$^+$ cells, and total populations were assessed with counting beads. **b** B memory cells, (**c**) B resident memory, (**d**) Isotype switched, (**e**) Plasma cells, (**f**) plasmablasts were reduced with sensory neuron ablation. Supplementation with VIP increased B cell populations. VIP supplementation did not further increase B cell populations in sensory neuron intact mice. **g** IgG was reduced with RTX as previously but increased with the supplementation of VIP to RTX mice. One-way ANOVA with Holm Sidak post hoc test, mean ± sd. Veh (PBS): $n = 8$, Veh (VIP): $n = 8$, RTX(PBS): $n = 7$, RTX(VIP): $n = 7$. **h** VIP1 receptor knockout (*VIPR1$^{-/-}$*) mice had reduced B memory cells compared WT mice,(**i**) B resident memory and, (**j**) Isotype switched B cells were similar in *VIPR1$^{-/-}$*and WT mice, (**k**) Plasma cells, and (**l**) plasmablasts were reduced in *VIPR1$^{-/-}$*vs WT mice. Two-sided *T* test. WT $n = 5$, VIPR1-/- $n = 5$. **m** *VIPR1$^{-/-}$*mice had reduced IgG following pre-exposure and infection compared to WT littermates. **n** VIP supplementation suppressed bacterial burden in RTX mice. No effect of VIP supplementation was observed in Veh mice. Veh (PBS): $n = 8$, Veh (VIP): $n = 7$-8, RTX(PBS): $n = 6$-7, RTX(VIP): $n = 6$-7. **o** *VIPR1$^{-/-}$*mice have increased bacterial burden following pre-exposure and infection with *S. pneumoniae* compared to WT littermates. One-way ANOVA with Holm Sidak post hoc test, mean ± sd. Data from 3 independent experiments. WT: $n = 11$, *VIPR1$^{-/-}$* $n = 7$. Two-sided t-test, mean ± sd. Data from 3 independent experiments. **p** Ly6g neutralizing antibody or isotype was given to WT or *VIPR1$^{-/-}$*mice; imagecreated in BioRender. Aguilar, D. (2022) BioRender.com/h64o602. Bacterial burden (Log CFU/ g tissue mass) from lung homogenates was greater with neutrophil depletion in WT mice compared to isotype control. *VIPR1$^{-/-}$* mice did not further increase bacterial burden with neutrophil depletion, demonstrating that VIP significantly stimulates B cells to increase neutrophil-mediated bacterial clearance. One-way ANOVA with Holm-Sidak post-hoc test, mean ± sd. WT(Iso): $n = 6$, WT(Ly6g): $n = 5$, *VIPR1$^{-/-}$*Iso): $n = 4$, *VIPR1$^{-/-}$* (Ly6g): $n = 5$. Data were pooled from two independent experiments.

that B cells clustered around neurons in the lungs and that the changes to B cell numbers were only in the lungs. Further, CD62L negative B resident cells were prominent in our model and reduced with sensory neuron ablation, demonstrating that tissue-resident B cells were employed upon infection after pre-exposure. In contrast to the lengthy time course required for germinal center-mediated B cell maturation, our rapid antibody responses were consistent with tissue-resident extra-follicular B cell antibody production[83,84]. We suggest that neuropeptides act to assist in the maturation and development of tissue-resident B cells.

Our data provide a new role of sensory neuropeptides acting as B cell stimulants and highlight the humoral influence of sensory neurons (Fig. 9). This study demonstrated that sensory neurons are critical in regulating pulmonary humoral immunity and the outcomes of bacterial lung infections and asthma. Targeting neuro-immunological communication through VIP and substance P or other potentially uncovered molecular mechanisms may be an effective approach to enhance host protection or reduce allergic responses by influencing immunoglobulin production through accessory pathways.

## Methods

### Mice
All animal experiments were approved by The Lundquist Institute at Harbor UCLA Institutional Animal Care and Use Committee protocol # 32183. Mice were housed in a specific-pathogen-free animal facility with 12:12 light: dark cycle, 21 ± 2 °C, ~ 40% humidity at The Lundquist Institute. C57BL/6 J #000664, *Tac1$^{-/-}$* (B6.Cg-Tac1tm1Bbm/J #004103), and μMT (*Ighm$^{-/-}$* B6.129S2-Ighmtm1Cgn/J #002288) mice were purchased from Jackson Laboratories. *Vglutcre-TdTomato* mice were provided by X. Sun (UCSD- crossed from *Vglut2cre-* and *RosaTdTomato-loxp* mice originally attained from JAX B6J.129S6(FVB)-Slc17a6tm2(cre)Lowl/MwarJ #028863 and B6.Cg-Gt(ROSA) 26Sortm14(CAG-tdTomato)Hze/J #007914). *TRPV1-DTR* mice were obtained from Dr. Isaac Chiu, with MTA provided by M Hoon (NIH)[51]. *VIPR1$^{-/-}$* were kindly donated by Joseph Pisegna, Patrizia Germano, and James Waschek (UCLA, MGI: 177616)[85]. At the start of experiments, mice were 5–8 weeks old. Age-matched male and female mice were used for experiments. Sensory neuron ablated mice (as below) were separated from vehicle/sham mice. Knockout mice were cohoused with WT littermates. At the end of each experiment, mice were deeply anesthetized (isoflurane 5% or ketamine/xylazine below) and exsanguinated for tissue harvest.

### Streptococcus pneumoniae cultures
*S. pneumoniae* strain 49619 serotype 19 F and strain 6303 serotype 3 were purchased from ATCC. Cultures were originally grown on Tryptic Soy Agar (BD Biosciences 236950) supplemented with 5% defibrinated sheep's blood (Remel R54012) and cultured for 24 h, then re-cultured for an additional 24 h in 5% CO$_2$ at 37 °C. Colonies were then picked for shape and hemolytic activity and cultured in Todd Hewitt Broth supplemented with 2% Yeast Extract (Fisher 50489152) or Brain Heart infusion media (BD DF0037178) in 5% CO$_2$ at 37 °C. Cultures were adjusted to OD 600 1.0 and resuspended in PBS prior to inoculation.

### Infection model
Colonies were checked for OD, then spun at 4000 rpm for 10 min, and washed in PBS. On day 0, mice were infected with 10$^4$ CFU of Serotype 19 F in 50 μl of PBS, intranasally. Then on day 9, mice were infected with 10$^8$ CFU of serotype 19 F in 50 μl of PBS intranasally. Bacterial burden was assessed in naïve mice (prior to any inoculation), immediately following the first inoculum (Day 1), prior to the second infectious dose (Day 8), 16 h after the infectious dose (Day 10), 48 h after the infectious dose (Day 11) and 6 days after the final infectious dose (Day 15). In a separate set of experiments, serotype cross-protection was tested. Mice were infected with 10$^4$ CFU of serotype 19 F in 50 μl of PBS, intranasally. Then on day 9, mice were infected with 10$^6$ CFU of serotype 3 in 50 μl of PBS intranasally., Survival and bacterial burden (at 48 h post-infection) were assessed. Control animals were intranasally infused with 50 μl PBS only.

### Allergy model
*A. alternata* extract was purchased from CiteQ Laboratories (09.01.26) and dissolved to yield a final concentration of 25 μg/ml. Then, mice were inoculated intranasally with 50ul on days 0, 3, 6, and 9 where experiments took place 16 or 48 h later[86]. All control mice received PBS in the same volume intranasally.

### Sensory neural ablation
For chemical ablation of TRPV1$^+$ neurons, mice 5 weeks of age were treated with RTX (Adipogen 502053716) as previously described[2,4,7]. Mice were injected subcutaneously in the dorsal flank on consecutive days with three increasing doses of RTX (30, 70, and 100 μg/kg) dissolved in 2% DMSO with 0.15% Tween 80 in PBS. Control mice received vehicle.

For targeted TRPV1+ neuron depletion, we injected 20 ng DT in 200 nl PBS containing 0.05% Retrobeads (LumaFluor INC) into nodose/jugular/petrosal VG with a nanoinjector (Drummond Scientific Company)[87]. Mice were anesthetized with 87.5 mg/kg ketamine and 12.5 mg/kg xylazine. The vagal ganglion was exposed after a midline incision in the neck (~ 1.5 cm in length). DT was gently injected at a rate of 5 nl/s. Once completed, the injection needle remained in the ganglia for 2-3 min in order to reduce the spillover of DT. This process was repeated for the vagal ganglion on the other side of the body.

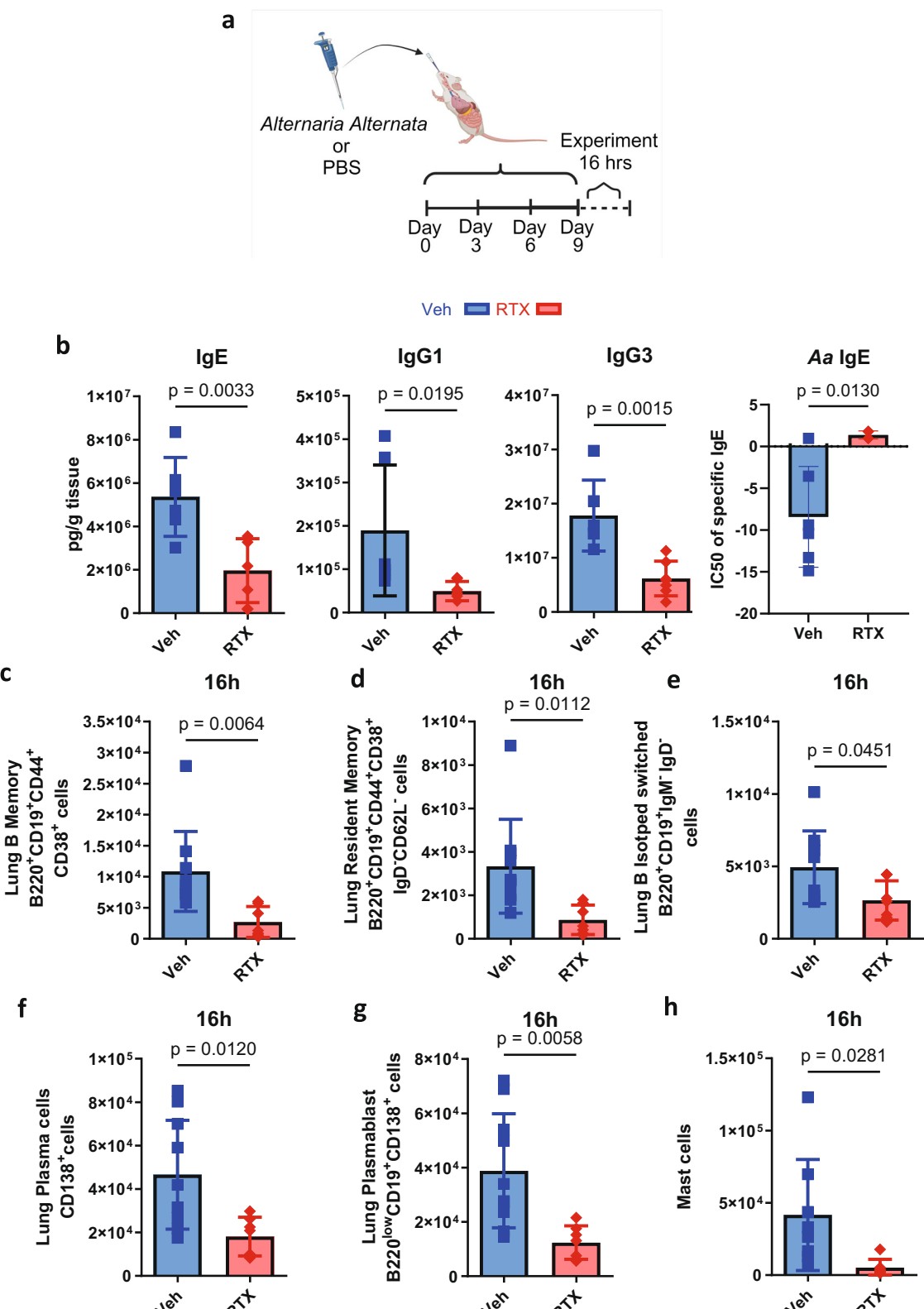

## Bacterial burden determination

Lungs were weighed and then homogenized in 2 ml sterile PBS with a tissue homogenizer. Homogenates were serially diluted on tryptic soy agar plates with 5% sheep's blood and 10ug/ml neomycin (Fisher Biochemicals BP2669-5). The bacterial CFU were enumerated after overnight incubation at 37 °C and 5% CO2.

## Neurotransmitter dosing

Vasoactive intestinal peptide (200 ng in 50 μl PBS) was delivered intranasally at 48, 24, and 0 h before and 24 h after (for 48 h harvest only) final infection with $10^8$ CFU of *S. pneumoniae*. Control mice received PBS at the same time points. When mice received bacteria on the same day, VIP was combined with the bacterial inoculum.

**Fig. 5 | Sensory neuron ablation reduces B lymphocytes, immunoglobulins, and mast cells in *A. alternata*-treated mice. a** *A. alternata* induced asthma was elicited as per Cavagnero et al.[86] 25 μg of *A. alternata* extract in 50 μl PBS was delivered as per the schematic image created in BioRender. Aguilar, D. (2022) BioRender.com/h64o602. Experiments took place 16 hrs later. **b** IgE, IgG1, and IgG3 (*n* = 6, RTX: *n* = 7) were reduced with sensory neuron ablation after the final *A. alternata dose*. Isotyping Luminex was used to analyze lung homogenates. *A. alternata* extract with lung homogenates and IC50 modeling was used to determine IgE specific to *A. alternata* (Veh *n* = 6, RTX *n* = 4) Two-sided *t* test. Data were

collected from two independent experiments. **c** B memory B220⁺CD19⁺CD44⁺CD38⁺ cells, (**d**) B resident memory B220⁺CD19⁺CD44⁺CD38⁺IgD⁻CD62l⁻ cells, (**e**) Isotype switched B220⁺CD19⁺IgM⁻IgD⁻ cells, (**f**) Plasma CD138⁺ cells and, (**g**) Plasmablasts CD138⁺B220^low CD19⁻ cells and (**h**) mast cells CD117⁺ were downregulated by sensory neuron ablation 16h after *A. alternata* asthma induction. Cells were gated on Live CD45⁺ cells, and counting bead calculation determined the total cell population. Veh: *n* = 10, RTX: *n* = 7. Two-sided *t* test, mean ± sd. Data from 3 independent experiments.

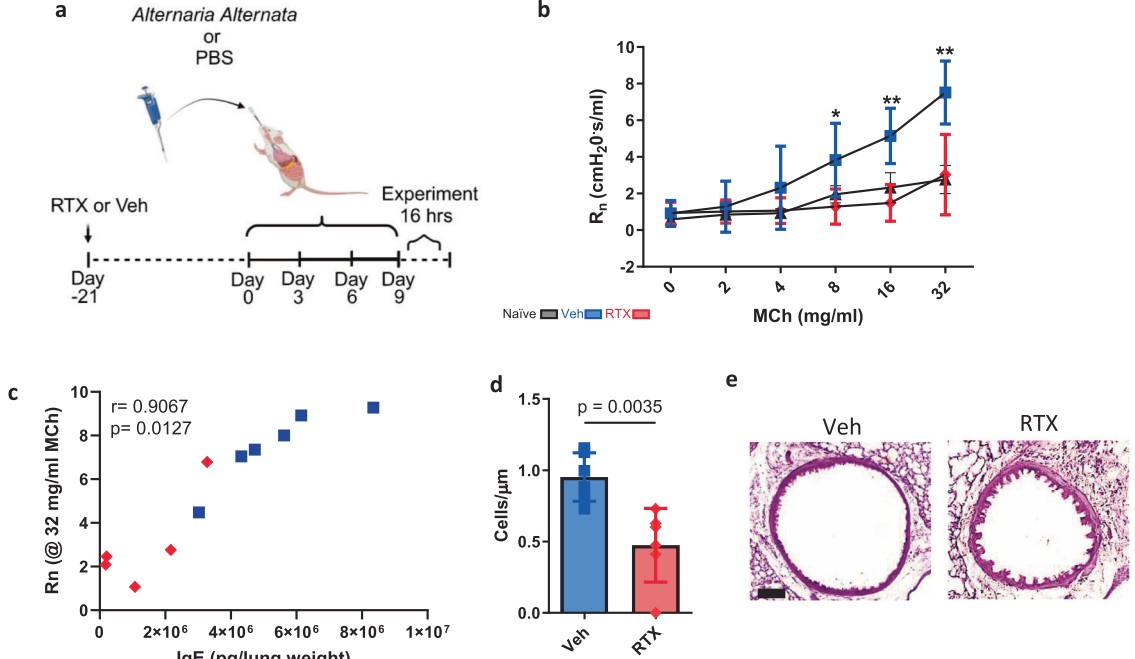

**Fig. 6 | Sensory neuron ablation reduces airway hyperresponsiveness and goblet cell metaplasia in *A. alternata-induced* asthma. a** *A. alternata-induced* asthma was elicited as per Cavagnero et al.[86] 25 μg of *A. alternata* extract in 50 μl PBS (PBS only for veh) was delivered as per the schematic image created in BioRender. Aguilar, D. (2022) BioRender.com/h64o602. Experiments took place 16hrs later. **b** RTX treated mice had a response to doubling doses of methacholine similar to PBS control mice and reduced compared to sensory neuron intact mice treated with *A. alternata*. PBS: *n* = 5, Veh: *n* = 6, RTX: *n* = 5. Two-way ANOVA with Tukey's

post hoc test. * Indicates a difference between Veh and RTX @8 mg/ml *p* = 0.0228. ** indicates difference from Veh and RTX 16&32 mg/ml: *p* = 0.0001, Veh and PBS 16 mg/ml: *p* = 0.0067, 32 mg/ml: *p* = 0.0001. Data from 2 independent experiments. **c** Correlation of IgE (from Fig. 5b) and Rn @ MCh (32 mg/ml) (**d**) Goblet cell metaplasia (Cells/μm) are reduced in RTX compared to sensory neuron intact *A alternata* treated mice. Veh: *n* = 6, RTX: *n* = 7.Two-sided *t* test, mean ± sd. Data from 2 independent experiments. **e** Representative images of periodic acid Schiff's reagent stain to visualize mucous producing Goblet cells. Scalebar = 20 μm.

Substance P (1000 ng in 50 μL PBS) was delivered intranasally on Days 5,6,7,8,9 of the *A. alternata* model. On days when *A. alternata* was delivered, and Substance P was mixed with *A. alternata*.

**Neutrophil depletion**
For neutrophil depletion, we followed an established protocol[88]. Mice were injected i.p. with 150 μg of anti-A18 (clone 1A8, BioXCell BE-0075-1) (in 200 μl) per mouse 48, 24, 0 h before lung infection and 24 h after lung infection (48 h analysis only). Control mice received 150 μg of rat IgG isotype control (2A3 BioXCell).

**Neutrophil bacteriocidal assay**
Mouse neutrophils were purified as described previously[89]. In brief, bone marrow cells were flushed from femurs and tibias of 8-week-old C57BL/6 J mice using sterile RPMI 1640 medium supplemented with 10% FBS and 2 mM EDTA onto a 50 mL screw top Falcon tube fitted with a 100 μm filter. Mouse neutrophils were purified from bone marrow cells using negative magnetic bead selection (MoJo Sort 480057, BioLegend) according to the manufacturer's instructions. Bone marrow-enriched neutrophils had >98% purity and >93%

viability. Neutrophil killing of S. *pneumoniae* 19 F was determined by CFU enumeration. *S. pneumoniae* 19 F was incubated with 50% decomplemented (20 min; 65 °C) serum in PBS from naïve, primary infected, and primary infected mice treated with RTX for 30 min on ice. Bacteria were washed with PBS, and $1 \times 10^3$ serum-coated *S. pneumoniae* 19 F were incubated with $1 \times 10^5$ BM-neutrophils for 30 min. Neutrophils were lysed with 0.02% Triton X-100 in ice-cold water for 5 min, diluted, and the remaining bacterial cells were quantitatively cultured.

**Flexivent lung function**
Mice anesthetized with ketamine/xylazine (87.5/12.5 mg/kg), paralyzed with pancuronium bromide (MP Biomedicals ICN15605350), and subject to methacholine challenge with 200 μl of doubling doses of methacholine (MP Biomedicals ICN19023110) dissolved in PBS for 20 breaths. Single and double compartment model Respiratory resistance was measured with the Flexivent system (SCIREQ) as previous[87]. Both absolute and percentage change values from PBS were calculated. Upon completion, mice were euthanized by anesthetic overdose (ketamine/xylazine) and subsequent exsanguination.

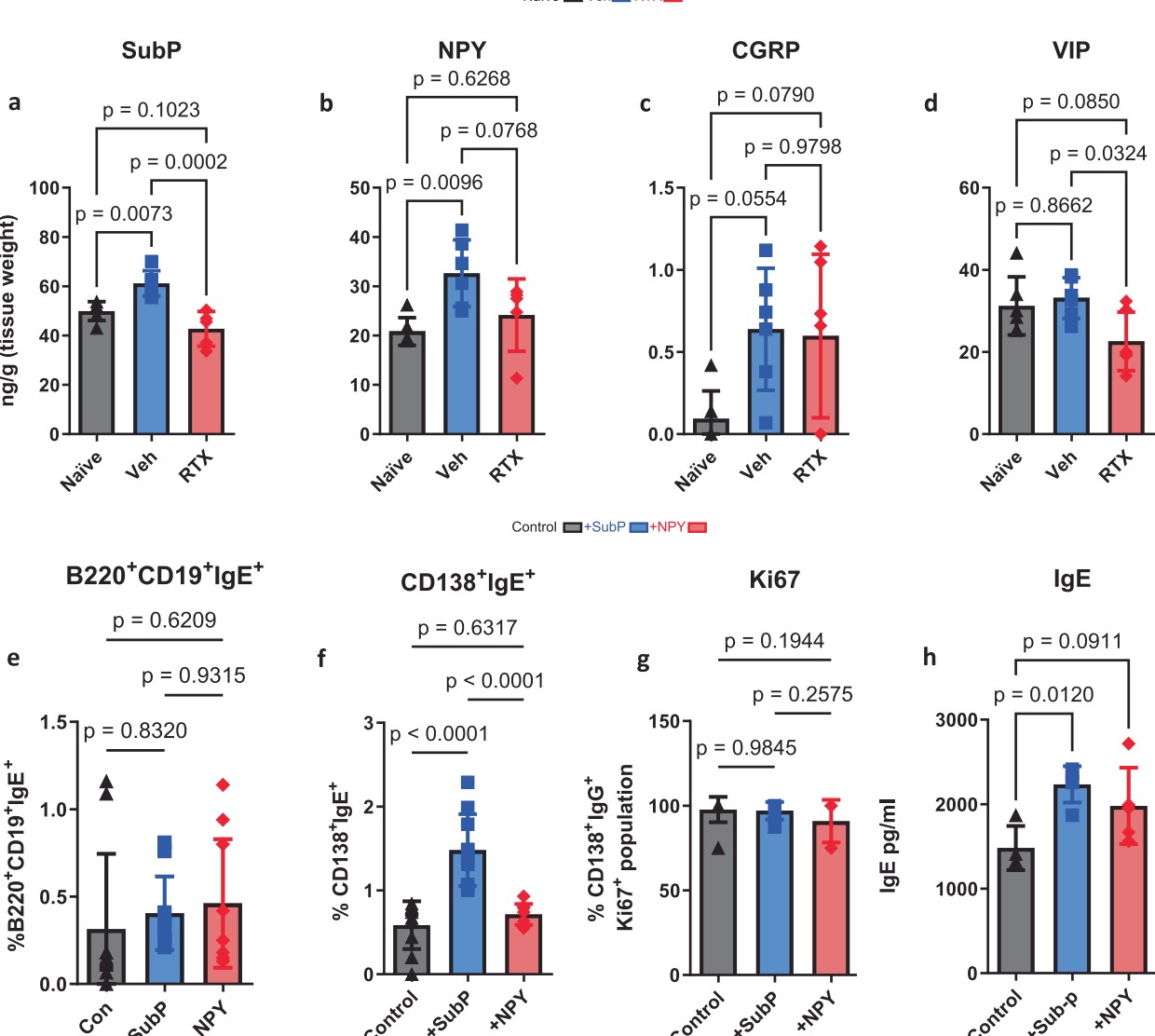

**Fig. 7 | *A. alternata* increases Substance P which augments IgE production.** *A. alternata* increased lung concentrations of Substance P (**a**) and NPY (**b**). CGRP (**c**) and VIP (**d**) were not significantly increased following *A. alternata* induction. Substance P (**a**) and VIP (**d**) were significantly reduced with RTX treatment. Analyzed from lung homogenates with ELISA. PBS: *n* = 6, Veh: *n* = 6, RTX: *n* = 6. One-way ANOVA with Tukey's post-hoc test. Data from 3 independent experiments. **e**–**h** B cells were isolated from spleens of naïve mice and cultured in media with IL4 + LPS and supplemented with Substance P (SubP) or NPY. Cells and media were sampled after 96 h of incubation. **e** Substance P did not increase bound IgE on B220+CD19+ cells. **f** Substance P significantly increased bound IgE in CD138+ plasma cells. **g** Proliferation, as assessed by intracellular Ki67 staining, was not affected by neuropeptides (from CD138+ cells). **h** Secreted IgE was increased by Substance P preferentially. IL4 + LPS: *n* = 4, +SubP: *n* = 5, + NPY: *n* = 5. One-way ANOVA and Tukey's post-hoc test, mean ± sd. Data from 2 independent experiments.

## Tissue collection

For flow cytometry, mice were euthanized by anesthetic overdose with isoflurane and subsequent exsanguination to collect organs. The lungs were then dissected and flushed with PBS, coarse dissected, and incubated at 37 °C for 45 min in 1.75 mg/ml collagenase IV (C4-22-1g Sigma) in PBS; then washed, macerated through a 21-gauge needle and filtered through a 70 μm mesh filter, then treated for flow cytometry. Spleens were macerated through a 70 μm mesh filter and then treated for flow cytometry. Bone marrow was flushed with RPMI media supplemented with 2 mM EDTA and 10% FBS, filtered through a 70 μm mesh filter, then treated for flow cytometry.

For cytokine analysis, lungs were dried, weighed, and then macerated in 2 ml PBS supplemented with 25 μl protease inhibitors (consisting of AEBSF HCl (100 mM), Aprotinin (80 μM), Bestatin (5 mM), E-64 (1.5 mM), Leupeptin (2 mM) and Pepstatin (1 mM) Tocris 5500), spun at 8000 g 4 min 4 C and then the supernatant was snap frozen and stored at −80 °C until analysis.

## B cell isolation

B cells were isolated from spleens or lungs (below) using the EasySep Mouse B cell negative selection kit. CD138 + cells (for qPCR, below) were isolated using the EasySep Mouse CD138 positive selection kit.

## Quantitative real-time PCR

RNA was isolated from tissues or cells using the inTron Easy spin Total RNA extraction Kit (Boca Scientific 17221), which was reverse transcribed to cDNA with a Tetro cDNA Synthesis Kit (Tetro biosystems

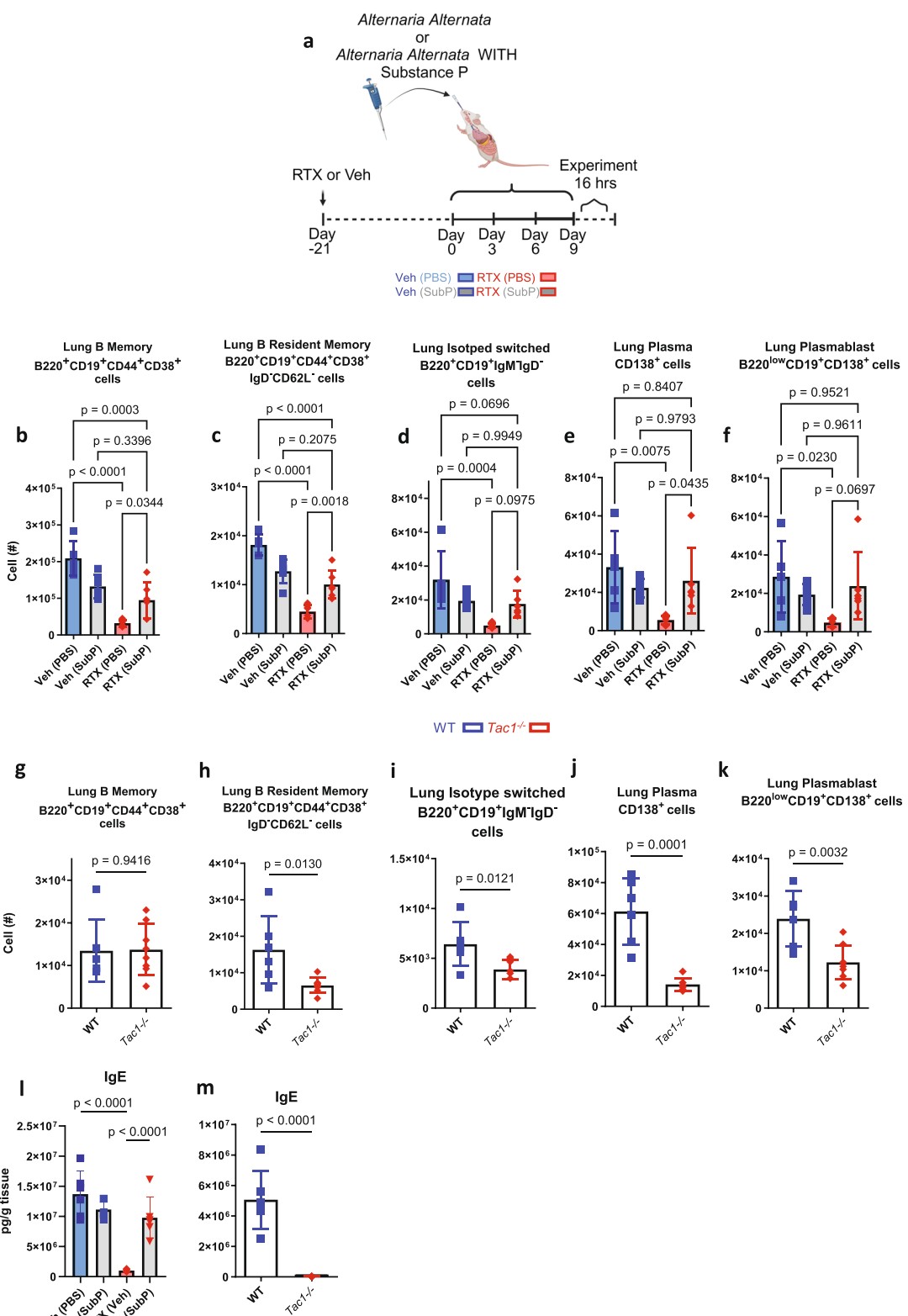

NC1352749). Relative gene expression was determined by quantitative real-time PCR on a QuantStudio 3 System (Thermo Fisher Scientific) with TaqMan Fast Advanced Master Mix (Thermo Fisher Scientific 4444557) with the TaqMan probe sets (ThermoScientific Applied Biosystems). Expression values relative to HPRT were calculated by Δ ΔCTT method. All taqman probes utilized were purchased from Thermofisher *Vipr1* Mm00449214_m1; *Vipr2* Mm01238618_g1; *Tac1r*

Mm00436892_m1; *Tac2r* Mm01175997_m1; *NPY1r* Mm00650798_g1; *NPY2r* Mm01956783_s1; *NPY4r* Mm00435894_s1; *NPY5r* Mm02620267_s1; *NPY6r* Mm00440546_s1; *MRGPRA1* Mm01984314_s1; *MRGPRB2* Mm01956240_s1; *MRGPRG* Mm01701870_s1; *RAMP1* Mm00489796_m1; *Tac1* Mm01166996_m1; *Calca* Mm00801463_g1; *VIP* Mm00660234_m1; *NPY* Mm01410146_m1; *TRPV1* Mm01246300_m1; *HPRT* Mm03024075_m1).

**Fig. 8 | Substance P increases B cells and IgE in mice treated with *A. alternata*.** **a** Substance P or vehicle (PBS) was delivered in conjunction with *A. alternata* extract as per the schematic, in accordance with our *A. alternata* model of asthma, the image created in BioRender. Aguilar, D. (2022) BioRender.com/h64o602. Cells were gated on Live/CD45 + cells, and total populations were assessed with counting beads. **b** B memory cells, (**c**) B resident memory, (**d**) Isotype switched, (**e**) Plasma cells, (**f**) plasmablasts were reduced with sensory neuron ablation. Supplementation with Substance P increased B cell populations. Substance P supplementation did not further increase B cell populations in sensory neuron intact mice. Veh (PBS): *n* = 5, Veh (Sub P): *n* = 6, RTX(PBS): *n* = 6, RTX(SubP): *n* = 6. One-way ANOVA with Holm sidak post hoc test, mean ± sd. Data from 3 independent experiments.

*Tac1*⁻/⁻ (lack Tac1 gene, unable to produce substance P, have receptors) had similar B memory (**g**), but reduced B Resident Memory (**h**), Isotype switched (**i**), plasma cells (**j**), and plasmablast cell (**k**) compared to WT mice receiving *A. alternata* induction of asthma. WT *n* = 6, *Tac1*⁻/⁻ *n* = 8, Two-sided *t* test, mean ± sd. Data from 2 independent experiments. **l** Supplementation with substance P increased IgE in *A. alternata* mice treated with RTX to a level consistent with Veh. Substance P did not further increase IgE in Veh mice (All groups *n* = 6). One-way ANOVA with Holm sidak post hoc test, mean ± sd. Data from 2 independent experiments. **m** *Tac1*⁻/⁻ (*n* = 7) had reduced IgE compared to WT *A. alternata* mice (*n* = 8). Two-sided *t* test, mean ± sd. Data from 2 independent experiments.

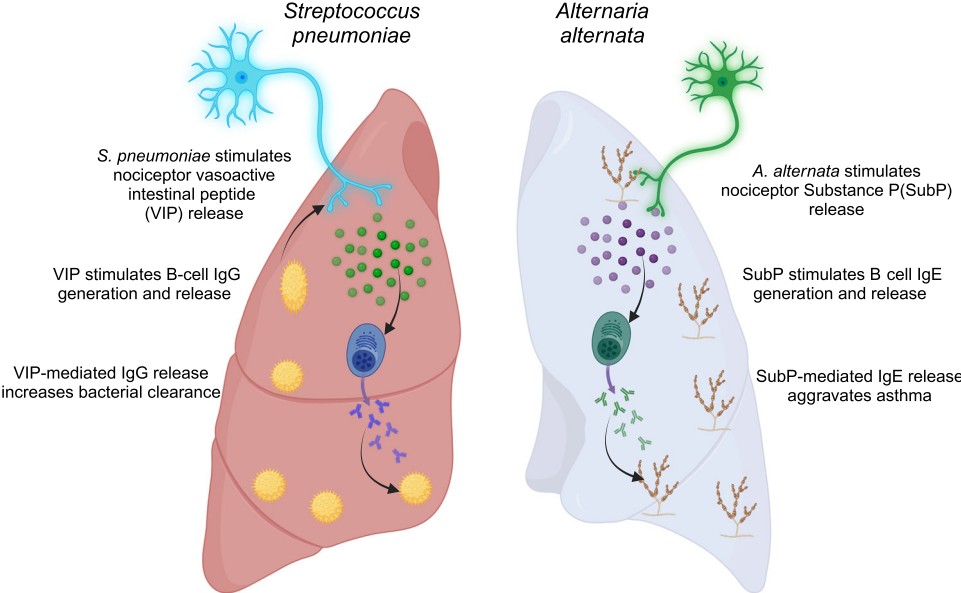

**Fig. 9 | Graphical summary. Sensory neurons stimulate B cells by the release of select neuropeptides, which enhance immunoglobulin production.** Sensory neuron neuropeptide release assists with infection but, exacerbates allergy. Image created in BioRender. Aguilar, D. (2022) BioRender.com/h64o602.

## Cytokine, neurotransmitter, and immunoglobulin analysis

Cytokine and immunoglobulin levels in lung homogenates were measured through Luminex multiplex assay using R&D discovery or Thermo kits; analytes were measured with the Luminex 200 system. Enzyme-linked immunosorbent assay (ELISA) kits were used according to the manufacturer's instructions (R&D systems, Raybiotech, or Thermofisher) and analyzed with a BioTek Synergy H1 plate analyzer. Luminex and ELISA kits are all commercially available and listed in the supplementary materials.

## Specific immunoglobulin analysis

For specific IgG determination, crude S. pneumoniae (12.53 µg/ml ATCC 49619) extract was incubated in 96 well plates overnight at 4 °C, then blocked with goat serum for 1 h at 4 °C. Then log dilutions of lung homogenates (from groups above) were incubated at RT for 2 h. Goat anti-mouse IgG HRP conjugated secondary antibody (Abcam ab205719, 1:1000) was then added and incubated for 2 h RT. The ELISA was developed with 0.5% TMB solution (Fisher AAJ61325AP) and stopped with 2 N H2SO4 (Fisher 828016). For specific IgE determination, *A. alternata* (100 µg/ml CiteQ 09.01.26) was incubated as above and developed as above, except Goat anti-mouse IgE HRP secondary (Thermofisher PA1-84764, 1:1000) was used. OD450 for each dilution was used with a 4-point sigmoidal curve to determine IC50 for each sample in each group.

## Flow cytometry

Red blood cells were lysed with ACK lysing buffer (Gibco A10492-01), treated with Fc Block (Biolegend 101320), and resuspended in FACS buffer (HBSS-Gibco 10010-023 with 2% FBS Gibco. Incubations with antibody cocktails were conducted at 4 °C for 60 min, and samples were subjected to two washes and resuspension in FACS buffer (HBSS + 2% FBS Sigma F8192). For intracellular staining, cells were fixed/permeabilized with BD cytofix/cytoperm kit (554714), washed, and stained overnight at 4 °C. Flow cytometry was conducted on a Symphony A5 flow cytometer (BD). Data were collected with BD DIVA software, and files were analyzed with FlowJo (Treestar, version 10.0.8r1). A live-cell stain (APC-Cy7, Invitrogen) was used to exclude dead cells. Gating strategies are provided in Supplementary Figs. Positive staining and gates for each fluorescent marker were defined by comparing full stain sets with fluorescence minus one (FMO) control stain sets. All antibodies were used at 1:100. Antibodies used: Biolegend: CD11c (Clone N418), Ly6g (Clone 1A8), CD117 (Clone 2B8), Siglec F (Clone 1RNM44N Invitrogen), F4 80 (Clone BM8), CD3 (BUV805), CD3 (Clone17As), CD11c (Clone N48), CD4 (Clone GK1.5), CD8 (Clone SK1), Tbet (Clone 4B10), GATA3 (Clone 16E10A23), B220 (Clone RA3-6B2), CD19 (Clone 6D5), CD62L (Clone MEL-14), CD44 (Clone IM7), CD138 (Clone 281-2), IgG1 (Clone RMG1-1), IgE (Clone RME-1) CD45 (Clone 30-F11), MRGPRX (Clone K125H4); BDbiosciences-IgM (Clone IL/41), IgD (Clone 11-26 C.2a), RoRγT (Clone Q31378; Invitrogen- CD38 (Clone 90), Ki67 (Clon SolA15); AlomoneLabs- VIPR1 (Clone AB_2341081), Tac1 (polyclonal Proteintech), VIP (polyclonal Proteintech).

All in vivo cells were gated on leukocytes, singlets, live, CD45⁺. Mast cells were CD117⁺MHCII⁺. Neutrophils were Ly6g⁺, Eosinophils were SiglecF⁺CD11b⁺. T cells were divided into CD4⁺CD8⁻ fractions and intracellularly stained for Tbet (Th1), or RORγT (γδ). B cell lineage was

B220$^+$CD19$^+$. These were subdivided based on B Memory (CD44$^+$CD38$^+$), B Resident Memory (CD44$^+$CD38$^+$IgD$^-$CD62L$^-$), and Isotype switched (IgM$^-$IgD$^-$). Plasma cells were CD138$^+$, plasmablasts were CD138$^+$B220$^{low}$CD19$^-$.

For assessment of either intracellular neuropeptides or neuropeptide receptors, cells were assessed from CD45$^+$, live singlets, and VIP or SP was stained intracellularly on CD3$^+$, CD117$^+$, Siglec F$^+$, Ly6G$^+$, or F480$^+$ cells. VIPR1 and MRGPRA1 were assessed on B220$^+$CD19$^+$ or CD138$^+$ cells.

For in vitro cultures, cells were divided into B220$^+$CD19$^+$ or CD138$^+$. Then, based on these gates assessed for IgG1 or IgE positivity. Ki67 was assessed from either CD138$^+$IgG1$^+$ or CD138$^+$IgE$^+$ gates. Exhaustion was assessed as CD11c$^+$CD23$^-$ from CD138$^+$ and B220$^+$CD19$^+$ gates.

### Immunofluorescence, histology, and microscopy

The lungs were perfused with PBS for immunostaining, followed by 4% paraformaldehyde (PFA) in PBS. Lungs were dissected and postfixed overnight in 4% PFA/PBS at 4 °C, incubated at 4 °C with 30% sucrose/PBS for two days, and stored in 0.1% sodium azide in PBS until cutting. Lungs were embedded in optimal cutting temperature compound (OCT, Tissue-Tek, PA), and 50-μm cryosections were cut at −20 °C and then blocked for 4 h in PBS with 10% donkey serum, 2% bovine serum albumin (BSA) and 0.8% Triton X-100. Sections were immunostained with the following antibodies: B220 (Biolegend 103225; 1:100 FITC), CD45 (Invitrogen 14045182 1:200), E-cadherin (Thermoscientific MA512547 1:200), VIP (Thermoscientific PA578224, 1:200), Substance P (Thermoscientific PA5106934 1:200) with appropriate secondary antibodies (Goat anti mouse Cy3 Jackson Immuno 115-165-003; Goat anti rat Alexa 647 Jackson Immuno 112-605-003; Goat anti rabbit Alexa 488 Abcam AB150077), washed and mounted with prolong antifade diamond (Thermofisher P36961).

Vagal (nodose/jugular) and dorsal root ganglia were dissected, fixed in 4%PFA for 1 h, cryopreserved in 30% sucrose overnight (all 4 C), then placed in OCT, and 14 μm sections were cut and placed on gelatin-coated slides. Sections were stained with TRPV1 (Alomone labs Clone-AB_2313819 1:200) followed by incubation with secondary antibody and secondary (Goat Anti Rabbit IgG H&L, Abcam). All sections were mounted with a Prolong diamond antifade compound (Thermofisher P36961).

For histology, lungs were embedded in paraffin and cut on a microtome (Leica) 5 μm and stained with hematoxylin (Epredia 6765001) and eosin (Epredia 6766007), gram stain (Sigma HT90T-1KT) or periodic acid, Schiff's reagent (Sigma395B-1kit). Blood smears were stained with Giemsa solution (AB150670). In addition, to confirm neutrophil counts in select experiments, Giemsa stain (Abcam AB150670) was used on lysed blood smears. Histological slides were mounted with permount toluene solution.

Sections were either imaged with a Leica SP8 confocal microscope or a Leica Thunder system. Data were imaged and analyzed with Leica LASx software. For quantification of the proximity of B cells to nerves, LASx online calipers were used to measure the distance from cell-to-nerve in 3D from the z-stack.

### B cell culture

Spleens or lungs were used to isolate B cells under sterile conditions. B cells were isolated using the EasySep Mouse B cell separation kit (StemCell Technologies 19854 A). Cells were then treated in B cell culture media made using 438 mL RPMI 1640 (Gibco 11835-030), 2.5 mL 1 M HEPES buffer (Sigma H3662), 5 mL GlutaMAX (Thermo Fisher Scientific 35050061), 5 mL penicillin/streptomycin stock (Sigma P0781), 30 mL heat-inactivated fetal bovine serum stock (Gibco A38400-01), and 454 μL β-mercaptoethanol (Gibco 21985-023) and vacuum filter sterilized (0.22 μm filter)[90]. Cells were then treated with

either media + IL4 (20 ng/ml, Sigma Aldrich I1020-5UG) + LPS (10 μg/ml, L6529-1MG Sigma Aldrich), media+IL4 + LPS + NPY (50 ng/ml, Tocris 1153), media+IL4 + LPS + VIP (50 ng/ml, Tocris 1911) or, media +IL4 + LPS+Substance P (50 ng/ml, 1156 Tocris). After 96 h media was stored at −80 °C until immunoglobulin analysis. Cells were stained and analyzed with a Symphony A5 flow cytometer, and data were processed with FlowJo (Treestar 10.881).

### Sample size and statistical analysis

We used animal numbers between 5 and 12 mice per experimental group/genotype for bacterial burden studies. For measurement of cytokine, neuropeptide, and immunoglobulins, 5-8 mice per group/genotype were used. For flow cytometry, 5–10 mice per group/genotype were used. For in vitro experiments, at least 3 animals were pooled into one sample, and 5–7 replicates were used. For sample measurements, 1 sample per animal per tissue was taken. Bacterial burden data were analyzed with the two-way RM ANOVA (time x group) with either Tukey or Holm-Sidak posthoc test; FACS and cytokines were compared with ANOVA with Tukey or Holm-Sidak posthoc tests, two-tailed unpaired t-tests for parametric analyses, or Mann–Whitney test for nonparametric analyses. Correlations were run with Pearson correlation. All statistical analyses were tested for normality. Data were plotted in Prism (GraphPad). All data are presented as mean ± sd.

### Reporting summary

Further information on research design is available in the Nature Portfolio Reporting Summary linked to this article.

### Data availability

The authors declare that the data supporting the findings of this study are available within the paper and the accompanying supplementary information files. Correspondence and requests for materials should be addressed to N.J. Source data are provided in this paper.

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

## Acknowledgements

NJ is supported by NIH grants R21AI159221, R56AI175328, UCLA CTSI UL1TR001881-01, and T32KT4708 of the Regents of the University of California Tobacco-Related Diseases Research Program. T.A.D. is supported by NIH AI171795 and Veterans Affairs BLR&D BX005073. N.M. and D.A. are supported by a California Institute for Regenerative Medicine Stem Cell Biology Training Grant EDUC4-12837. The content is solely the responsibility of the authors and does not necessarily represent the official views of the National Institutes of Health. We are grateful to Dr. Isaac Chiu (Harvard) for providing *TRPV1-DTR* mice as per the direction of Dr. Mark Hoon (NIH). We are grateful to Dr. Xin Sun for providing *Vglutcre-Tdtomato* mice. Figures 1a, 2a, 2i, 4a, 4p, 5a, 6a 8a, 9 and Supplementary Figs. 2a were made using Biorender and are used in this manuscript on a CC-BY-NC-ND license.

## Author contributions

D.A. and F.Z. contributed to the experimental design, conducted experiments, and analyzed data. A.M. and N.M. contributed to experimental execution and data analysis. P.G., J.P., T.A.D., and O.A. assisted with experimental design and analysis. M.S. contributed to the study design and manuscript preparation. N.J. designed the study, conducted experiments, analyzed data, and prepared the manuscript and figures. All authors agree on the manuscript.

## Competing interests

D.A., M.S., and N.J. declare the following competing interests. U.S. Patent Application Serial Number 63/492,846. Methods of using sensory neuron neurotransmitters to enhance humoral immunity. The remaining authors do not declare competing interests.
