## [Peer Review File · Nature Communications]

Sensory neurons regulate stimulus-dependent humoral immunity in mouse models of bacterial infection and asthmaREVIEWER COMMENTS

Reviewer #1 (expert in fungal asthma):

Summary:

The study highlights the crucial role of sensory neurons in coordinating immune responses, particularly in the context of humoral immunity within the lung. It demonstrates that sensory neurons are essential for the recruitment of B-cells and plasma cells, as well as antibody production, in response to *Streptococcus pneumoniae* infection and allergen-induced airway inflammation. The findings reveal that different neuropeptides are released by sensory neurons depending on the type of immunogen, suggesting that targeting sensory neurons could offer new treatment strategies for diseases related to impaired or excessive humoral immunity.

Major:

1. The *Streptococcus pneumoniae* colonization model requires further elucidation. The methodology for determining the appropriate dose and timing for this infection model is not clearly defined. A pivotal aspect that remains unaddressed is the impact of sensory neuron depletion on mortality rates in a lethal bacterial infection model. Given that the present observations ceased at 48 hours, the long-term effects of neuron depletion on bacterial colonization are ambiguous. Questions persist regarding whether the bacterial load will ultimately diminish or remain persistent, and the timeline for such outcomes needs clarification.
2. In the *Alternaria*-induced asthma model, while data on cellularity, immunoglobulins, and cytokine levels are comprehensive, outcomes directly related to asthma, such as lung function and mucous metaplasia, are inadequately presented. Specifically, in Supplemental Figure 11, the anomalous lung resistance observed at the baseline in *Alternaria*-challenged mice suggests potential lung tissue damage or airway obstruction. The observation that methacholine challenges failed to significantly alter resistance levels except at the highest concentration implies a lack of reversible airway hyperresponsiveness. The reduction in baseline resistance following RTX treatment further raises questions about whether RTX confers protection against injury from fungal proteases. Additionally, the observation of denuded airway epithelium in RTX-treated samples prompts concerns about the potential cytotoxic effects of RTX on airway epithelial cells.
3. It is acknowledged that both Vasoactive Intestinal Peptide (VIP) and Substance P are synthesized by a variety of non-neuronal cells, including neuroendocrine and immune cells. The data presented does not fully explore the contribution of these alternative sources, nor does it adequately discuss their potential roles. It is plausible that supplementing these neuropeptides could activate non-neuronal pathways, thereby contributing to the restoration of lung homeostasis. This aspect warrants a more thorough investigation to fully understand the complex roles and interactions of VIP and Substance P within the lung, beyond their neuronal origins.

Minor:

1. The manuscript exhibits several grammatical and structural inaccuracies that necessitate correction. For instance, the term "immune stimulus-dependent" on Line 72 could be more precisely articulated as "immune-stimulus-dependent" to clarify the dependency relation. Furthermore, the sentence on Lines 135-137 appears incomplete and lacks coherence. It could be revised for clarity, perhaps to something like, "To investigate humoral immunity in

response to *S. pneumoniae* lung infection, we aimed to replicate the colonization and infection patterns observed in humans, considering that mice are not naturally colonized with *S. pneumoniae*."

2. The labels for Figure 2c-d are obscured.
3. The *Alternaria* treatment scheme is absent from Figure 5.
4. There is a discrepancy regarding Figure 3h and its legend; the figure is missing, yet its legend is present, creating confusion. Conversely, the legend for Figure 3i is absent.
5. The legend for Figure 4i has been incorrectly labeled as Figure 4j.

Reviewer #2 (expert in respiratory medicine):

Very interesting and novel (to me at least) data showing sensory neurones are necessary for an effective B cell response to *S. pneumoniae* and subsequent protection against infection. This was mediated by VIP. In related data for an asthma model they show that the sensory neurones potentiate asthma through B cell production of IgE and this was dependent on Substance P not VIP.

However, although the data linking sensory neurones and VIP to B cells and antibody mediated immunity are strong, the mechanism involved was unclear. There are several areas where further clarity is needed as follows:

1. For text only; presumably the VIP comes from the neurones themselves and signals to the B cells? How does *S. pneumoniae* stimulate VIP production by a neurone? is this direct (bacteria interaction with neurone) or indirect (cytokine or PAMP mediated??)
2. One important thing is whether the B cell numbers are altered in mice depleted of sensory neurones by RTX before *S. pneumoniae* infection? We really need to know as this is a clear potential mechanism that could underpin most of their findings, rather than changes in B cell populations after infection.
3. Ex vivo B cell splenocyte expts in Fig 3 are supportive of a role of VIP in influencing B cell function; but these data would be better if they were obtained for lung B cells obtained from *S. pneumoniae* infected animals.
4. The above feeds into the mechanism question which is unclear to me at present; it seems there is both reduced B cell numbers in neurone depleted mice and that VIP stimulates B cell function. Which is the mechanism the authors favour and why? The B cell numbers alone would cause the increased susceptibility phenotype.
5. The VIPR KO data are useful, but are incomplete compared to the data for the RTX treated mice; the paper would be strengthened by having a full set of data for these mice eg B cell subpopulations and cytokines along with the existing Cfu data)
6. The data for uMT, neutrophil depleted, and VIP supplemented mice are all useful and supportive but really do not link the loss of neurones to each of these; experiments combining B cell depletion / uMT / neutrophil depletion with RTX or VIPR KO; or passive transfer of antibody into RTX treated would help link the neurone / B cell neutrophil access together.

Minor issues

Text changes for clarity / sense needed here and there eg lines 135-38 196-98, 222-319
what were the timepoints for Fig 2A and 2I
Fig 6 panel e is obscured by panel h

Reviewer #3 (expert in neuroimmunology):

This study delineates the critical role of sensory neurons in modulating humoral immunity, especially in the lung's response to *Streptococcus pneumoniae* and allergic asthma. Sensory neurons facilitate B-cell and plasma cell recruitment and antibody production. Depleting sensory neurons increases bacterial burden and diminishes immune response to *S. pneumoniae*, whereas it reduces B-cell populations, IgE levels, and asthmatic symptoms in allergic asthma models. The research further reveals that different stimuli prompt sensory neurons to release distinct neuropeptides: vasoactive intestinal polypeptide (VIP) for bacterial infections and substance P for asthma. Manipulating these neuropeptides affects the immune response, indicating that sensory neurons play a versatile role in immune coordination and represent a potential target for treating diseases with dysregulated humoral immunity.

The work is interesting, well-written, and carefully executed. However, a few points need to be clarified before publication.

MAJOR POINTS

1. Regarding SF10: We note with surprise the absence of TACR1/TACR2 expression in B cells. Instead of analyzing naive spleen cells, could you investigate the lung or draining lymph nodes (dLN) B cells from infected or allergic mice and compare them to naive ones? We suggest measuring neuropeptide receptor expression across different B cell populations, including B220+/CD19+ or CD138+ cells.
2. Regarding Figure 2: The manuscript presents circulating immunoglobulins and B cell polarization levels, but it remains to be seen if you have data on immunoglobulin-producing B cells in the lung. Please clarify this.
3. Concerning Figure 2h: The data at the 16-hour mark raises the question of how mice lacking B cells (uMT) exhibit lower *S. pneumoniae* colony-forming units (CFU). Could you explain this phenomenon? One would expect the opposite.
4. Regarding Figure 3f/h: Would it be possible for you to test the effects of calcitonin gene-related peptide (CGRP) and substance P (SP) and culture B cells and repeat the NPY/VIP experiments with cultured B cells from mice lacking *Npy1r* or *Vipr1* genes?

MINOR POINTS

5. For Figure 3f/h: Could you include fluorescence-activated cell sorting (FACS) plots related to these findings?
6. About Figures 4g-4j: We request a correlation analysis between CFUs, circulating peptide levels, and IgG levels. Could you provide this analysis?
7. Regarding Figure 6h, the legends and plots regarding TackO are somewhat confusing.

Could you clarify that this pertains to Tac1 (gene encoding for SP) germline knockout mice?

8. Regarding SF1: It would be insightful to show that RTX also reduces VIP and SP levels in jugular and nodose ganglion (JNC) neurons.

9. In SF11: Linking the findings to mucus metaplasia or airway hyperreactivity (AHR) by correlating lung IgE/SP levels or testing the impact of rescuing SP in the lungs of RTX-treated mice would add value to the study.

10. Line 100: Please note that reference ten has shown similar findings in vivo.

11. Line 184 Clarification: We presume you intended to refer to "Vglut2cre-td-tomato." Could this be confirmed?

12. The authors should consider adding discussion on AHR and Mucus Metaplasia: Considering the known link between SP and mucus metaplasia, incorporating discussion supported by references such as PMIDs: 33580593, 1716217, 31954778, 26119026, 19381016, which could enrich the manuscript.

13. Clarification on Peptide Rescue experiments: In the legends of Figures 4g/l and Figure 6 and the result section, could you clarify when and how peptides were rescued in these mice and whether this was done using a peptidase inhibitor, given the sensitivity of these neuropeptides to enzymatic degradation?

We thank the reviewers for their positive and constructive comments. We have addressed each query below. Our responses are in red text. Please note that we indicate where we have provided additional text/discussion (in the **MARKED** revised manuscript), and in which figure we have added additional data. For convenience, we also place the new data in line with our responses.

NCOMMS-24-08353 R1 comments

Reviewer 1.

1. The *Streptococcus pneumoniae* colonization model requires further elucidation. The methodology for determining the appropriate dose and timing for this infection model is not clearly defined. A pivotal aspect that remains unaddressed is the impact of sensory neuron depletion on mortality rates in a lethal bacterial infection model. Given that the present observations ceased at 48 hours, the long-term effects of neuron depletion on bacterial colonization are ambiguous. Questions persist regarding whether the bacterial load will ultimately diminish or remain persistent, and the timeline for such outcomes needs clarification.

Our doses were established in our lab with consultation from Dr. Mizgerd and his vast knowledge of *S. pneumoniae* infection (PMID: 28513594, 34060477, 38715601). After 2 doses of 19F, Etesami et al. found that resident B cells were expanding 1 day after the second dose of 19F. Therefore, we note that our increase in B-cells are resident to the lungs, similar to PMID:38715601. To further define our model, we have additional data alongside data presented in the initial manuscript. Our methods have been described in greater detail (Lines: 416-435), and Figure 1 along with the results section (Lines: 142-168), have been revised accordingly.

We show that RTX and Veh naïve mice have no *S. pneumoniae* in their lungs prior to infection (Naïve mice- in 1st submission), the day after the first 'low' inoculum, bacteria is present (Day 1), before the infectious dose bacteria is cleared in both Veh and RTX mice (Day 8), Day 10 (16 hours- in 1st submission) Vehicle and RTX mice have a similar bacterial burden, Day 11 (48 hours- in 1st submission) RTX mice are unable to clear bacteria as well as vehicle mice, and 6 days after the infectious dose on Day 15 ,the burden remains high in RTX versus vehicle mice.

Mice received escalating doses of resiniferatoxin (RTX) or Vehicle (Veh) starting 21 days prior to pre-exposure. 10^4 CFU of *S. pneumoniae* (Serotype 19F ATCC49619) in 50 μ l PBS was delivered on day 0 to expose mice, then 10^8 CFU in 50 μ l PBS was delivered on Day 9. Bacterial burden was enumerated before any infection (naïve) after pre-exposure (Day 1), before infection (Day 8), after infection (16h-Day 10, 48h-Day 11) and 6 days after (Day 15). Veh: n=5-13, RTX: n=5-7. Two-way ANOVA with Newman-Keuls post hoc test. Data were pooled from three independent experiments.

We further conducted cross-strain-protection experiments with serotype 19F as the first pre-exposure/inoculum and then used strain 6303 (serotype 3), a lethal strain, for infection. Specifically, the first dose was the same 10^4 CFU of strain 19F in 50 μ l PBS; then, however, we gave 10^6 CFU of serotype 3 in 50 μ l PBS on day 9 and assessed survival and bacterial burden 48h later. Like inoculation with 19F and infection with 19F, when inoculated with 19F and infected with 6303, RTX mice cleared fewer bacteria from their lungs compared to Vehicle mice.

10^4 CFU of *S. pneumoniae* (Serotype 19F ATCC49619) in 50 μ l PBS was delivered on day 0 to expose mice, then 10^6 CFU (Serotype 3, ATCC6303) in 50 μ l PBS was delivered on Day 9, bacterial burden was assessed on Day 11. Two-sided t-test. Veh n=7, RTX n=6.

When we conducted a survival experiment following inoculation with 19F and infection with 6303, RTX mice were unable to resist the level of infection compared to Vehicle mice.

10⁴ CFU of *S. pneumoniae* (Serotype 19F ATCC49619) in 50μl PBS was delivered on day 0 to expose mice, then 10⁶ CFU (Serotype 3, ATCC6303) in 50μl PBS was delivered on Day 9, survival was assessed with Log-rank (Mantel-Cox) test. Veh, n=7, RTX, n=6.

In order to confirm that the clearance of 6303 was due to inoculation, we conducted a survival experiment following only a single dose of 6303 without pre-exposure. Similar to the data provided in our initial manuscript in supplementary Figure 2 showing without inoculation, Veh and RTX mice clear 19F similarly at 48h and have similar IgG, the survival following a single dose of 6303 was not different between Veh and RTX mice. This data is shown in Supplementary Figure 2d.

10⁶ CFU of *S. pneumoniae* (Serotype 3 ATCC6303) in 50μl PBS was delivered survival was assessed with Log-rank (Mantel-Cox) test. Veh n=7, RTX, n=6.

Given that no differences in bacterial burden were seen before the final infectious dose and that the protection offered by the initial pre-exposure is in place in response to two different strains of streptococcal infection at various timepoints, we conclude that sensory neurons play a role in host-defense to clear pulmonary *S. pneumoniae* infection by assisting humoral immune memory.

2. In the Alternaria-induced asthma model, while data on cellularity, immunoglobulins, and cytokine levels are comprehensive, outcomes directly related to asthma, such as lung function and mucous metaplasia, are inadequately presented.

Specifically, in Supplemental Figure 11, the anomalous lung resistance observed at the baseline in Alternaria-challenged mice suggests potential lung tissue damage or airway obstruction. The observation that methacholine challenges failed to significantly alter resistance levels except at the highest concentration implies a lack of reversible airway hyperresponsiveness. The reduction in baseline resistance following RTX treatment further raises questions about whether RTX confers protection against injury from fungal proteases. Additionally, the observation of denuded airway epithelium in RTX-treated samples prompts concerns about the potential cytotoxic effects of RTX on airway epithelial cells.

We identified issues with our Flexivent itself in previous airway hyperresponsiveness experiment, which have since been fixed. The repeat experiment is now shown (Supplementary figure 16b).

A. alternata induced asthma was elicited as per Cavagnero et al. 25µg of *A. alternata* extract in 50µl PBS (PBS only for veh) was delivered as per the schematic in supplementary figure 16a. Experiments took place 16 hrs later. RTX-treated mice had a response to doubling doses of methacholine similar to PBS control mice and reduced compared to sensory neuron intact mice treated with *A. alternata*. PBS: n=5, Veh: n=6, RTX: n=5. Two-way ANOVA with Tukey's post hoc test. * Indicates difference between Veh and RTX, p<0.01. ** indicates difference from Veh and RTX, Veh and PBS, p<0.01. Data from 2 independent experiments.

Our data demonstrate a hyperresponsiveness pattern similar to previous investigations (PMID: 29403029; Supplemental figure 16b). That RTX would provide 'protection' against fungal proteases to induce airway hyperresponsiveness is consistent with the notion that sensory airways instigate airway hyperresponsiveness in response to allergens. Trankner et al. (PMID: 25049382) were the first to demonstrate that TRPV1+ neurons are responsible for airway hyperresponsiveness; our data are consistent with this notion.

To the authors' knowledge, we have not seen data to suggest that RTX would damage the epithelium. RTX targets TRPV1+ cells. A wealth of data demonstrates that TRPV1 is confined to sensory neurons (see discussion in PMID: 21139565, 29505031). Therefore, only TRPV1+ sensory neurons would become apoptotic in response to RTX treatment. Nevertheless, we wanted to confirm that the epithelium is not adversely affected by RTX. We conducted immunohistochemistry for cleaved caspase 3 (a marker of apoptotic cell death). We show similar staining in Vehicle or RTX *A alternata* induced mice.

Cleaved Caspase = Green (FITC)
DAPI= Blue

Positive control mouse lung explants treated with hedgehog pathway inhibition increasing apoptosis

With regards to airway denudation, we note that what appears to be 'lifting' of the epithelium off the submucosa, is a blood vessel. We agree with the reviewer that this is not the ideal image. We provide all the images collected below for PAS staining and have replaced images in Supplemental figure 16e with new images. Of note, we did not detect a denudation of the epithelium in the RTX airways compared with vehicle controls.

3. It is acknowledged that both Vasoactive Intestinal Peptide (VIP) and Substance P are synthesized by a variety of non-neuronal cells, including neuroendocrine and immune cells. The data presented does not fully explore the contribution of these alternative sources, nor does it adequately discuss their potential roles. It is plausible that supplementing these neuropeptides could activate non-neuronal pathways, thereby contributing to the restoration of lung homeostasis. This aspect warrants a more thorough investigation to fully understand the complex roles and interactions of VIP and Substance P within the lung, beyond their neuronal origins.

We acknowledge possible alternate sources of neuropeptides. We have conducted additional experiments to address this issue. When Vglutcre-tdTomato mice (express tdtomato for sensory nerves under the Vglut gene) underwent our *S. pneumoniae* and *A. alternata* models (with and without RTX), we see that the VIP and SP fluorescence signal is higher in Vglut+ cells than those stained for CD45 (to capture all lymphocytes and granulocytes) or E-cadherin (staining the epithelium, where neuroendocrine cells would reside) in transverse lung sections. Furthermore, RTX only suppressed VIP and SP fluorescence in Vglut+ cells. These new data are shown in Supplementary Figure 15 (for *S. pneumoniae*) and Supplementary Figure 22 (for *A. alternata*).

S. pneumoniae

Vglutcre-tdtomato mice underwent our pre-exposure/infection *S. pneumoniae* model as per Figure 1a. Lungs were harvested and stained for CD45, E-cadherin and VIP. The VIP fluorescence signal was normalized to the CD45, E-cadherin or Vglut fluorescence signal. RTX only reduced VIP in Vglut expressing cells. Two-way ANOVA with Holm-Sidak post-hoc test. N=3 per group. Scalebar= 50 μ m.

A. alternata

Vglutcre-tdtomato mice underwent *A. Alternata* as per Figure 5a. Lungs were harvested and stained for CD45, E-cadherin and Substance P (SP). The SP fluorescence signal was normalized to the CD45, E-cadherin or Vglut fluorescence signal. RTX only reduced SP in Vglut expressing cells. Two-way ANOVA with Holm-Sidak post-hoc test. N=3 per group. Scalebar= 50µm.

When we stained for intracellular VIP or SP (these peptides would be stored in vesicles and not expressed on the cell surface) in T-cells, macrophages, mast cells, eosinophils, and neutrophils from lung single cell suspensions, RTX did not reduce this signal.

S. pneumoniae

Lungs from Veh or RTX mice which were subject to pre-exposure and infection to *S. pneumoniae* 19F were processed for flow cytometry. Cells were gated from CD45, live, singlets. RTX and Vehicle mice exposed to our pre-exposure and infection model had similar VIP expression in T-cells (CD3), Macrophages (F4/80) and neutrophils (Ly6g). N=8 per group. Two-sided t-test.

A. alternata

Lungs from Veh or RTX mice which were subject to *A. alternata* were processed for flow cytometry. Cells were gated from CD45, live, singlets. RTX and Vehicle mice exposed to *A. alternata* had similar SP expression in T-cells (CD3), eosinophils (Siglec F) and mast cells (CD117). N=8 per group. Two-sided t-test.

Finally, VIP and SP supplementation did not further augment VIP or SP in cells from either model.

S. pneumoniae

Veh and RTX mice underwent VIP supplementation or PBS along with pre-exposure and infection with *S. pneumoniae* 19F. Lungs were prepared for flow cytometry 48h after infection. Cells were gated from CD45, live, singlets. VIP expression in T-cells (CD3), Macrophages (F4/80) and neutrophils (Ly6g) were not different between groups N=7-8 per group, One-way ANOVA with Holm-Sidak post-hoc test. Two independent experiments.

A. alternata

Veh and RTX mice underwent SP supplementation or PBS along with *A. alternata* induction of asthma. Lungs were prepared for flow cytometry 48h after infection. Cells were gated from CD45, live, singlets. SP expression in T-cells (CD3), eosinophils (Siglec F) and mast cells (CD117) were not different between groups N=7-8 per group, One-way ANOVA with Holm-Sidak post-hoc test. Two independent experiments.

In conjunction with the measurement of these peptides from lung homogenates conducted initially, showing that RTX suppressed these neuropeptides, we conclude that the most significant source of these neuropeptides are the sensory neurons.

Nevertheless, it is possible that these alternate sources of neuropeptides may be stimulated in other ways; we also discuss this potential contribution (Lines: 254-265, 299-311, 333-341).

Appropriate antibodies were listed in methods, Lines 539-547.

Minor:

1. The manuscript exhibits several grammatical and structural inaccuracies that necessitate correction. For instance, the term "immune stimulus-dependent" on Line 72 could be more precisely articulated as "immune-stimulus-dependent" to clarify the dependency relation. Furthermore, the sentence on Lines 135-137 appears incomplete and lacks coherence. It could be revised for clarity, perhaps to something like, "To investigate humoral immunity in response to *S. pneumoniae* lung infection, we aimed to replicate the colonization and infection patterns observed in humans, considering that mice are not naturally colonized with *S. pneumoniae*."

We have corrected these suggestions and revised the manuscript's punctuation and small grammatical errors.

2. The labels for Figure 2c-d are obscured.

Figure 2 includes additional data. The legend has been reworked, and all labels correctly inserted.

3. The *Alternaria* treatment scheme is absent from Figure 5.

Added.

4. There is a discrepancy regarding Figure 3h and its legend; the figure is missing, yet its legend is present, creating confusion. Conversely, the legend for Figure 3i is absent.

Figure 3 includes additional data. The legend has been reworked, and all labels correctly inserted.

5. The legend for Figure 4i has been incorrectly labeled as Figure 4j.

Figure 4 includes additional data to satisfy reviewer concerns. The legend has been reworked, and all labels have been correctly inserted.

Reviewer #2 (expert in respiratory medicine):

Very interesting and novel (to me at least) data showing sensory neurones are necessary for an effective B cell response to *S. pneumoniae* and subsequent protection against infection. This was mediated by VIP. In related data for an asthma model they show that the sensory neurones potentiate asthma though B cell production of IgE and this was dependent on Substance P not VIP.

However, although the data linking sensory neurones and VIP to B cells and antibody mediated immunity are strong, the mechanism involved was unclear. There are several areas where further clarity is needed as follows:

1. For text only; presumably the VIP comes from the neurones themselves and signals to the B cells? How does *S. pneumoniae* stimulate VIP production by a neurone? is this direct (bacteria interaction with neurone) or indirect (cytokine or PAMP mediated??)

We added context in the introduction (Lines: 74-75) and discussion (Lines: 324-328). As the reviewer suggests, investigating how sensory neurons detect and produce neuropeptides is outside the confines of this study, but warrants an explanation to the reader.

2. One important thing is whether the B cell numbers are altered in mice depleted of sensory neurones by RTX before *S. pneumoniae* infection? We really need to know as this is a clear potential mechanism that could underpin most of their findings, rather than changes in B cell populations after infection.

We conducted additional experiments to analyze B-cells from lungs in Veh and RTX mice immediately before pre-exposure (Naïve), and immediately before infection (Day 8) to complement our '16h' (Day 10) and '48h' (Day 11) analysis. These data are now analyzed with the previous data using a Two-way (Day vs group) ANOVA and are in revised Figure2c-g, Lines 176-189.

B-memory B220+/CD19+/CD44+/CD38+ cells, B resident memory B220+/CD19+/CD44+/CD38+/IgD-/CD62l- cells, Isotype switched B220+/CD19+/IgM-/IgD- cells, Plasma CD138+ cells and, Plasmablast CD138+/B220-/CD19- cells were downregulated by sensory neuron ablation at 16 and 48h after infection with 19F, but not altered before pre-exposure and minimally altered before infection with 19F. Data from four independent experiments. Two-way ANOVA with Holm-Sidak test. Veh: n=5-7, RTX n=5-7.

These data show that RTX does not inhibit B-cell populations in the lungs of naïve mice. These data are also supported by the fact that global B-cells are similar between Veh and RTX mice in spleen and bone marrow (Supplementary Figure 7 from original submission). Further, lung B-cells are the same before pre-exposure, and most cell types are not different between Veh and RTX before infection. It appears that RTX mice cannot increase the number of B-cells or plasma cells in the lungs following infection (Figure 2c-g) and, therefore, cannot resist infection (Figure 1b-d).

3. Ex vivo B cell splenocyte exps in Fig 3 are supportive of a role of VIP in influencing B cell function; but these data would be better if they were obtained for lung B cells obtained from *S. pneumoniae* infected animals.

We repeated *in vitro* experiments with B-cells harvested from naïve mice and mice who underwent our pre-exposure and infection model of *S. pneumoniae*. We again saw that VIP increased bound and released IgG.

B-cells were harvested from the lungs of naïve and pre-exposed and infected *S. pneumoniae* mice (48h after final exposure). Culture conditions were the same with 96h of incubation with media (IL4+LPS), media +VIP, media+SP, media+NPY or media+CGRP. VIP consistently increased bound and released IgG. Two-way ANOVA with Holm-Sidak post-hoc. Naïve n=4-7, *S. pneumoniae* n=4-7.

However, to our surprise, the lung B-cells harvested from pre-exposed and infected *S. pneumoniae* lungs which were then stimulated in culture for 96h showed a blunted response. We reasoned that this was due to B-cell exhaustion. Indeed, when we checked the cultured lung B-cells for CD11c⁺/CD23⁻ expression as per PMID: 21543762, 21562046, 36072594, we saw a distinct rise in exhaustion in pre-exposed and infected lung B-cells with additional culture stimulation.

B-cells were harvested from the lungs of naïve and pre-exposed and infected *S. pneumoniae* mice (48h after final exposure). Culture conditions were the same with 96h of incubation with media (IL4+LPS), media +VIP, media+SP, media+NPY or media+CGRP. Pre-exposed and infected lung B-cells showed increased CD11c and reduced CD23 expression after additional culture stimulation. Two-way ANOVA with Holm-Sidak post-hoc. Naïve n=4-7, *S. pneumoniae* n=4-7.

We discuss these findings further (Lines 234-240) and present the data in Figure 3m-q.

4. The above feeds into the mechanism question which is unclear to me at present; it seems there is both reduced B cell numbers in neurone depleted mice and that VIP stimulates B cell function. Which is the mechanism the authors favour and why? The B cell numbers alone would cause the increased susceptibility phenotype.

The major effect that neuropeptides have on B-cells directly is via the stimulation of immunoglobulin production and class-switch. Overall, this conclusion is supported by Veh vs RTX, neuropeptide supplementation experiments, comparison of WT and knockout mice *in vivo*, *in vitro* experiments and *in vitro* experiment from WT and knockout mice.

5. The VIPR KO data are useful, but are incomplete compared to the data for the RTX treated mice; the paper would be strengthened by having a full set of data for these mice eg B cell subpopulations and cytokines along with the existing Cfu data).

We have expanded our analysis to include B-cell subsets, cytokines and all immunoglobulins in VIP1R^{-/-} vs WT and Tac1^{-/-} vs WT from Lung B-cells and lung homogenates. These new data are in Figure 4h-l, Supplementary Figure 14, Figure 6g-k and Supplementary Figure 21 respectively, Lines 241-253; 289-299.

S. pneumoniae VIP1R^{-/-} vs WT

B-memory B220⁺/CD19⁺/CD44⁺/CD38⁺ cells were reduced in VIP1R^{-/-} vs WT, B resident memory B220⁺/CD19⁺/CD44⁺/CD38⁺/IgD⁻/CD62L⁻ cells were similar between WT and VIP1R^{-/-} mice, Isotype switched B220⁺/CD19⁺/IgM⁻/IgD⁻ cells were similar between WT and VIP1R^{-/-} mice, Plasma CD138⁺ cells were lower in VIP1R^{-/-} vs WT mice and, Plasmablast CD138⁺/B220^{low}/CD19⁺ cells were downregulated in VIP1R^{-/-} mice. Two-sided t-test. WT: n=11 VIP1R^{-/-}: n=7.

VIP1R^{-/-} mice have reduced immunoglobulins, but not cytokines compared to WT mice following pre-exposure and infection to *S. pneumoniae*. Quantification of BAFF, IL-1β, IL-4, IL-5, IL-13, IL-33, CCL5, CCL7, CXCL13, TNFα, IgA, IgM, IgG1, IgG2a, IgG2b, IgG3, IgE in WT and VIP1R^{-/-} following pre-exposure and infection to *S. pneumoniae* WT: n=6-8, VIP1R^{-/-}: n=5-7. Two-sided t-test. Data from 2 independent experiments.

A. alternata Tac1^{-/-} vs WT

Tac1^{-/-} (lack Tac1 gene, unable to produce substance P, but have receptors) had similar B memory, but reduced B Resident Memory, Isotype switched, plasma cells, and plasmablast cell compared to WT mice receiving *A. alternata* induction of asthma. WT: n=6, Tac1^{-/-}: n=8, Student's unpaired t-test. Data from 2 independent experiments.

Tac1^{-/-} mice have reduced inflammatory, b-cell survival, recruiting cytokines, chemokines, Th2 cytokines and immunoglobulins following *A. alternata* treatment. Quantification of BAFF, IL-1β, IL-4, IL-5, IL-13, IL-33, CCL5, CCL7, CXCL13, TNFα, IgA, IgM, IgG1, IgG2a, IgG2b, IgG3, IgE (repeated from figure 6) in Wild type and Tac1^{-/-} mice after the final dose of *A. alternata*. WT: n=7-8, Tac1^{-/-}: n=7-8. Two-sided t-test. Data from 2 independent experiments.

6. The data for uMT, neutrophil depleted, and VIP supplemented mice are all useful and supportive but really do not link the loss of neurones to each of these; experiments combining B cell depletion / uMT / neutrophil depletion with RTX or VIPR KO; or passive transfer of antibody into RTX treated would help link the neurone / B cell neutrophil access together.

We conducted new experiments to deplete neutrophils in VIP1R^{-/-} vs WT mice. These data show that neutrophil depletion in VIP1R^{-/-} mice does not further exacerbate bacterial burden compared to VIP1R^{-/-} mice given the isotype antibody; unlike WT mice where neutrophil neutralization exacerbated bacterial burden. These new data are shown in Figure 4p (Lines 249-253).

Ly6g neutralizing antibody or isotype was given to WT or VIP1R^{-/-} mice. Bacterial burden (Log CFU/g lung weight) from lung homogenates was greater with neutrophil depletion in WT mice compared to isotype control. VIP1R^{-/-} mice did not further increase bacterial burden with neutrophil depletion demonstrating that VIP significantly stimulates B-cells to increase neutrophil-mediated bacterial clearance. One-way ANOVA with Holm-Sidak post-hoc test. WT(Iso): n=6, WT(Ly6g): n=5, VIP1R^{-/-}(Iso): n=4, VIP1R^{-/-}(Ly6g): n=5. Data were pooled from two independent experiments.

In conjunction with the neutrophil depletion assay in RTX vs Veh mice, the reduction of IgG in RTX vs Veh mice and VIP1R^{-/-} vs WT mice and further additional data showing that decompartmented serum from Veh mice allows enhanced neutrophil killing of strain 19F compared to decompartmented serum from RTX (Figure 2j, Lines 198-202), these data show that VIP stimulates B-cells to increase immunoglobulin release to stimulate neutrophil mediated bacterial killing (Methods: Lines-467-478).

Naïve neutrophils were incubated with decompartmented serum harvested from naïve, Veh (pre-exposed and infected) or RTX (pre-exposed and infected) mice and plated in *S. pneumoniae* 19F coated wells (5000 CFU). Neutrophils were lysed 30 minutes later and plated on blood agar plates and CFU were enumerated 24h later. One-way ANOVA with Holm-Sidak post-hoc, n=6 per group.

Minor issues

Text changes for clarity / sense needed here and there eg lines 135-38 196-98, 222-319

Corrected.

what were the timepoints for Fig 2A and 2I

We have added the appropriate timepoints in the figure legend.

Fig 6 panel e is obscured by panel h

Corrected.

Reviewer #3 (expert in neuroimmunology):

This study delineates the critical role of sensory neurons in modulating humoral immunity, especially in the lung's response to *Streptococcus pneumoniae* and allergic asthma. Sensory neurons facilitate B-cell and plasma cell recruitment and antibody production. Depleting sensory neurons increases bacterial burden and diminishes immune response to *S. pneumoniae*, whereas it reduces B-cell populations, IgE levels, and asthmatic symptoms in allergic asthma models. The research further reveals that different stimuli prompt sensory neurons to release distinct neuropeptides: vasoactive intestinal polypeptide (VIP) for bacterial infections and substance P for asthma. Manipulating these neuropeptides affects the immune response, indicating that sensory neurons play a versatile role in immune coordination and represent a potential target for treating diseases with dysregulated humoral immunity.

The work is interesting, well-written, and carefully executed. However, a few points need to be clarified before publication.

MAJOR POINTS

1. Regarding SF10: We note with surprise the absence of TACR1/TACR2 expression in B cells. Instead of analyzing naive spleen cells, could you investigate the lung or draining lymph nodes (dLN) B cells from infected or allergic mice and compare them to naive ones? We suggest measuring neuropeptide receptor expression across different B cell populations, including B220+/CD19+ or CD138+ cells.

We isolated CD138+ cells and global B-cells using the EasySep Mouse CD138+ selection and EasySep Mouse B-cell isolation kit, respectively, from the lungs of naïve mice, lungs of mice following our *A. alternata* model and lungs of mice following our *S. pneumoniae* model. We conducted qPCR for: NPY1R, NPY2R, NPY4R, NPY5R, NPY6R, TAC1R, TAC2R, VIP1R, VIP2R, MRGPRA1, MRGPRB2, MRGPRG and RAMP1. Intriguingly, VIP1R, VIP2R and RAMP1 were genes with lowest $2^{\Delta CT}$ values in all groups and cell types showing the highest expression. We note that Tac receptors were undetectable in spleen but present in lung B-cells and CD138+ cells. There were no differences between groups. We note however, that VIP1R and RAMP1 were the highest expressed in all groups and cells from both tissues. These new data are shown in supplementary figure 11 (Methods: Lines-496-498, 506-511; Results: Lines-224-226, 282-287).

Lung B-cell Control vs SP19F

Spleen B-cell Control vs SP19F

Lung CD138 Control vs SP19F

Spleen CD138 Control vs SP19F

Lung B-cell Control vs *Alternaria Alternata*

Spleen B-cell Control vs *Alternaria Alternata*

Lung CD138 Control vs *Alternaria Alternata*

Spleen CD138 Control vs *Alternaria Alternata*

CD138+ and B-cells were isolated from the lungs and spleens of naïve, *S. pneumoniae* and *A. Alternata* mice (n=5 mice were pooled per sample, 2 samples per group). Cells were isolated with the CD138 positive or B-cell negative EasySep selection kit. qPCR for prominent neuropeptide receptors was run, and data were expressed as ΔCT from the hypoxanthine-thymine-thymine housekeeping gene. No differences between groups within the respective subset were demonstrated for any receptor. Although we note that VIP1R, VIP2R, RAMP1 are highly expressed in all samples.

2. Regarding Figure 2: The manuscript presents circulating immunoglobulins and B cell polarization levels, but it remains to be seen if you have data on immunoglobulin-producing B cells in the lung. Please clarify this.

B-cell populations from Figure 2 were from lungs. In response to reviewer #2 we have added additional data to Figure 2c-g to show the progression of B-cells throughout our *S. pneumoniae* pre-exposure and infection model. Spleen and bone marrow cell populations are in supplementary figure 7 from the original manuscript. This has been clarified in the revised manuscript (Lines 176-189). Data in the *A. alternata* model were also from Lung B-cells (Figure 5, 6, original manuscript).

B-memory B220+/CD19+/CD44+/CD38+ cells, B resident memory B220+/CD19+/CD44+/CD38+/IgD-/CD62L- cells, Isotype switched B220+/CD19+/IgM-/IgD- cells, Plasma CD138+ cells and, Plasmablast CD138+/B220-/CD19- cells were downregulated by sensory neuron ablation at 16 and 48h after infection. Data from four independent experiments. Two-way ANOVA with Holm-Sidak test. Veh: n=5-7, RTX: n=5-7.

3. Concerning Figure 2h: The data at the 16-hour mark raises the question of how mice lacking B cells (uMT) exhibit lower *S. pneumoniae* colony-forming units (CFU). Could you explain this phenomenon? One would expect the opposite.

Initially, we too, were intrigued by this finding. However, μ MT mice have documented neutrophilia (PMID: 36111205, 36178806, 10725726) and appear to be able to clear a number of acute infections. Therefore, μ MT are able to clear bacteria initially but are overwhelmed and less able to clear as time goes on, due to an inefficient clearing of bacteria as they are not identified by immunoglobulins. We further demonstrate that IgG signaling to neutrophils is required for streptococcal clearance as serum, which was de complemented, harvested from pre-exposed and infected mice stimulate greater neutrophil phagocytic capabilities than de complemented serum from non-pre-exposed mice and RTX mice (Figure 2j. Methods: Lines-467-478; Results: Lines-198-202).

Naïve neutrophils were incubated with decompemented serum harvested from naïve, Veh or RTX pre-exposed and infected mice and plated in *S. pneumoniae* 19F coated wells (5000 CFU). Neutrophils were lysed 30 minutes later and plated on blood agar plates and CFU were enumerated 24h later. One-way ANOVA with Holm-Sidak post-hoc, n=6 per group.

Therefore, μ MT mice, like RTX mice are unable to produce sufficient immunoglobulins to augment neutrophil killing capacity.

4. Regarding Figure 3f/h: Would it be possible for you to test the effects of calcitonin gene-related peptide (CGRP) and substance P (SP) and culture B cells and repeat the NPY/VIP experiments with cultured B cells from mice lacking *Npy1r* or *Vipr1* genes?

We have cultured B-cells from WT and *VIP1R*^{-/-} mice. We show that *VIP1R*^{-/-} B-cells do not respond to VIP and produce less IgG when stimulated with VIP, SP, NPY and CGRP. These data are now shown in Figure 3j-l, Lines 231-234.

B-cells were isolated from the spleens of *VIP1* receptor knockout (*VIP1R*^{-/-}) and WT littermates and cultured for 96h. *B220*/*CD19*⁺ cells were not significantly affected by *VIP1R* deletion. *VIP1R*^{-/-} *CD138*⁺ cells did not significantly upregulate IgG in response to VIP and other neuropeptides, whereas WT *CD138*⁺ cells were stimulated by VIP and other neuropeptides. IgG released into the media was not increased with VIP or other peptides with *VIP1R* genetic deletion; VIP increased WT B-cell IgG release. Two-way ANOVA with Holm-Sidak post-hoc test. N=4-7. Control = grey, VIP = blue, SP = orange, NPY = red, CGRP = green.

MINOR POINTS

5. For Figure 3f/h: Could you include fluorescence-activated cell sorting (FACS) plots related to these findings?

We have added flow plots in supplementary figure 12.

6. About Figures 4g-4j: We request a correlation analysis between CFUs, circulating peptide levels, and IgG levels. Could you provide this analysis?

We have run the correlation matrix between CFUs, peptide levels in lung homogenates and immunoglobulin levels from lung homogenates. Peptides and immunoglobulins were significantly inversely correlated with lung CFUs. Peptides and immunoglobulins were significantly positively correlated. These are shown in supplementary figure 10 (Lines 217-222).

Pearson R

P-values

	CFU	NPY	SubP	CGRP	VIP	IgA	IgM	IgE	IgG1	IgG2a	IgG2b	IgG3
CFU		0.01853	0.04710	0.24975	0.00438	0.03758	0.02219	0.06014	0.00045	0.08170	0.00667	0.01060
NPY	0.01853		0.00326	0.02953	0.00320	0.00762	0.08230	0.04060	0.00180	0.04487	0.00194	0.00013
SubP	0.04710	0.00326		0.00130	0.00142	0.00022	0.06640	0.25111	0.01760	0.33736	0.01195	0.06704
CGRP	0.24975	0.02953	0.00130		0.06616	0.00577	0.07582	0.97280	0.02450	0.50163	0.21350	0.12286
VIP	0.00438	0.00320	0.00142	0.06616		0.00030	0.03329	0.31057	0.00884	0.27689	0.03333	0.04195
IgA	0.03758	0.00762	0.00022	0.00577	0.00030		0.02699	0.60501	0.00341	0.38203	0.02330	0.09246
IgM	0.02219	0.08230	0.06640	0.07582	0.03329	0.02699		0.80710	0.00605	0.02400	0.22491	0.05849
IgE	0.06014	0.04060	0.25111	0.97280	0.31057	0.60501	0.80710		0.18249	0.05487	0.00080	0.02906
IgG1	0.00045	0.00180	0.01760	0.02450	0.00884	0.00341	0.00605	0.18249		0.03256	0.00545	0.00066
IgG2a	0.08170	0.04487	0.33736	0.50163	0.27689	0.38203	0.02400	0.05487	0.03256		0.05679	0.00159
IgG2b	0.00667	0.00194	0.01195	0.21350	0.03333	0.02330	0.22491	0.00080	0.00545	0.05679		0.00709
IgG3	0.01060	0.00013	0.06704	0.12286	0.04195	0.09246	0.05849	0.02906	0.00066	0.00159	0.00709	

Lung bacterial burden (Figure 1b), immunoglobulins (Figure 2a) and neuropeptides (Figure 3b, c, d, e) were compared with Pearson correlation. CFU was inversely correlated with neuropeptides and immunoglobulins with VIP and IgG1 showing the strongest relationships to Lung bacterial burden (CFU). Neuropeptides and immunoglobulins were positively correlated with one another.

7. Regarding Figure 6h, the legends and plots regarding TackO are somewhat confusing. Could you clarify that this pertains to Tac1 (gene encoding for SP) germline knockout mice?

We have clarified the figure legend and text (Line 293-299).

8. Regarding SF1: It would be insightful to show that RTX also reduces VIP and SP levels in jugular and nodose ganglion (JNC) neurons.

We have run qPCR on prominent neuropeptides in the nodose/jugular ganglion in vehicle and RTX mice (Supplementary figure 1; Lines 137-138).

The prominent neuropeptides- vasoactive intestinal peptide (VIP), substance P (denoted by Tac1 gene), calcitonin gene related peptide (denoted by CALCA gene) and neuropeptide Y (NPY) were all down-regulated in the nodose/jugular ganglion with RTX treatment). Two-sided t-test of ΔCT values. Housekeeping gene = hypoxanthine phosphoribosyltransferase. N=3 mice per sample (6 ganglia per sample), N=3 samples per group and condition.

9. In SF11: Linking the findings to mucus metaplasia or airway hyperreactivity (AHR) by correlating lung IgE/SP levels or testing the impact of rescuing SP in the lungs of RTX-treated mice would add value to the study.

We provide the correlation between IgE and MCh (32mg/ml) in supplementary figure 16c and added to the results section (Line 272-275).

Correlation of IgE (from Figure 5b) and Rn @ MCh (32mg/ml) (from Supplementary figure 16b). Red = RTX, Blue = Veh.

10. Line 100: Please note that reference ten has shown similar findings in vivo.

We have revised this statement and included reference 10 (Lines 104-105 in revised manuscript).

11. Line 184 Clarification: We presume you intended to refer to "Vglut2cre-td-tomato." Could this be confirmed?

Confirmed. We have corrected this (Lines 207 in revised, marked manuscript).

12. The authors should consider adding discussion on AHR and Mucus Metaplasia: Considering the known link between SP and mucus metaplasia, incorporating discussion supported by references such as PMIDs: 33580593, 1716217, 31954778, 26119026, 19381016, which could enrich the manuscript.

We have expanded our discussion to include these references (Lines 375-380).

13. Clarification on Peptide Rescue experiments: In the legends of Figures 4g/l and Figure 6 and the result section, could you clarify when and how peptides were rescued in these mice and whether this was done using a peptidase inhibitor, given the sensitivity of these neuropeptides to enzymatic degradation?

We did not use peptidase inhibitors. The most common peptidase inhibitor used to prevent neuropeptide cleavage is dipeptidyl peptidase 4. However, this substance is used to increase the amount of GLP-1 for the management of type II diabetes (PMID: 24068868). Given the presence of GLP-1 in the lung and that GLP-1 has profound effects in the lung (PMID: 37491292), we did not want to have off-target effects with our rescue/supplementation experiments.

We note that both VIP and substance P are cleaved into peptide fragments and that these peptide fragments bind to VIP and Substance P receptors, respectively (PMID:35336804, 2654011,1711051,24382888, 21946672, PMC3273185, 26336928); once bound the effects are long-lasting. Therefore, the binding of peptides and their fragments to receptors on B-cells mediate immunoglobulin release.

As a supplement, we measured VIP in lung homogenates of Veh and RTX mice subject to pre-exposure and infection with *S. pneumoniae* 19F with and without VIP supplementation (Supplementary figure 13), Lines 241-244.

Vasoactive intestinal peptide supplementation increases vasoactive intestinal peptide in RTX mice. Mice were treated as per Figure 4a. VIP was measured from lung homogenates and expressed per lung weight. VIP supplementation significantly increased VIP levels in lungs of RTX mice 16hrs after final infectious dose of bacteria. Veh (PBS): n=6, Veh (VIP): n=7, RTX(PBS): n=5, RTX(VIP): n=6. One-way ANOVA with Holm sidak post hoc test. Data from 3 independent experiments.

and substance P in Veh and RTX mice subject to *A. alternata* with and without substance P supplementation (Supplementary figure 20, Lines 287-289).

Substance P supplementation increases Substance P in RTX mice. Mice were treated as per Figure 6a. Substance P was measured from lung homogenates and expressed per lung weight. Substance P supplementation significantly increased Substance P levels in lungs of RTX mice 16hrs after final *A. alternata* dose. Veh (PBS): n=6, Veh (SubP): n=6, RTX(PBS): n=6, RTX(SubP): n=5. One-way ANOVA with Holm sidak post hoc test. Data from 3 independent experiments.

These data show that neuropeptides are increased with supplementation, which would suggest minimal degradation of the neuropeptides in the lungs.

REVIEWERS' COMMENTS

Reviewer #1 (Remarks to the Author):

All my comments have been addressed in the revised manuscript.

Reviewer #2 (Remarks to the Author):

The authors have added a substantial amount of new data that in my opinion addresses the issues raised by myself and the other reviewers

Reviewer #3 (Remarks to the Author):

It was great to see that the authors did a tremendous amount of work on the revision. All my concerns have been addressed, and the paper is now acceptable for publication in NC.

We have provided our point-by-point response to reviewers concerns below, in red text.

REVIEWERS' COMMENTS

Reviewer #1 (Remarks to the Author):

All my comments have been addressed in the revised manuscript.

Excellent. Thank you for your helpful comments.

Reviewer #2 (Remarks to the Author):

The authors have added a substantial amount of new data that in my opinion addresses the issues raised by myself and the other reviewers

Wonderful. We appreciate the your time and efforts as a reviewer.

Reviewer #3 (Remarks to the Author):

It was great to see that the authors did a tremendous amount of work on the revision. All my concerns have been addressed, and the paper is now acceptable for publication in NC.

Fantastic. Thank you for your kind comments.